

# The evolution of snow bedforms in the Colorado Front Range and the processes that shape them

Kelly Kochanski[1,2,3], Robert S. Anderson[1,2], and Gregory E. Tucker[1,3]

[1]Department of Geological Sciences, University of Colorado at Boulder, Boulder, CO 80309, USA
[2]Institute of Arctic and Alpine Research, University of Colorado at Boulder, Boulder, CO 80309, USA
[3]Cooperative Institute for Research in Environmental Sciences, University of Colorado at Boulder, Boulder, CO 80309, USA

**Correspondence:** Kelly Kochanski (kelly.kochanski@colorado.edu)

**Abstract.** When wind blows over dry snow, the snow surface self-organizes into bedforms such as dunes, ripples, snow-waves, and sastrugi. These bedforms govern the interaction between wind, warmth, and the snowpack, but thus far they have far attracted few scientific studies. We present the first time-lapse documentation of snow bedform movement and evolution, as part
of a series of detailed observations of snow bedform movement in the Colorado Front Range. We show examples of the movement of snow ripples, snow-waves, barchan dunes, snow-steps, and sastrugi. We also introduce a previously undocumented bedform: the stealth dune. These observations show that (1) snow dunes accelerate minute-by-minute in response to gusts; (2) sastrugi and snow-steps present steep edges to the wind, and move by retreating downwind; (3) snow-waves and dunes deposit layers of cohesive snow in their wakes; and (4) bedforms evolve along complex, cyclic trajectories. We use these observations to
build for new conceptual models of bedform evolution, based on the relative fluxes of snowfall, aeolian transport, erosion, and snow sintering across and into the surface. We find that many snow bedforms are generated by complex interactions between these processes. The prototypical example is the snow-wave, in which deposition, sintering, and erosion occur in transverse stripes across the snowscape.

## 1  Introduction

Wind-blown snow self-organizes into bedforms, such as dunes, waves, and ripples, the most common of which are anvil-shaped sastrugi. These bedforms cover sea ice, tundra alpine ridges, and almost all of Antarctica (Filhol and Sturm, 2015). They are formed by the interaction of wind, heat, and snow. After they form, bedforms govern the interactions between the snow surface and the atmosphere.

Snow bedforms increase the variability of snow depth on short (0.1–5 m) length scales. This variability increases the uncer-
tainty on paleoclimate measurements of snow accumulation from ice cores (Leonard (2009) §2.5.3), and decreases the average thermal conductivity of the snow (Liston et al., 2018), while focusing heat flux in areas of thin snow (Petrich et al., 2012). Bedforms also increase the aerodynamic drag of the surface (Inoue, 1989a, b; Jackson and Carroll, 1978; Amory et al., 2017), and change the distribution of wind-blown aerosols (Harder et al., 2000). Finally, they increase the surface area of the snow,



which decreases its reflectivity (Leroux and Fily, 1998; Warren et al., 1998; Corbett and Su, 2015). All of these effects are difficult to measure remotely.

Snow bedforms are commonly 0.1–5 m long. They are generally sub-pixel-scale for satellite imagery, and definitively sub-grid-scale for global and regional climate models. Moreover, bedforms align with the direction of the prevailing and/or most recent wind (Amory et al., 2016). Their effects on aerodynamic roughness and reflectivity are therefore anisotropic (Leroux and Fily, 1998; Corbett and Su, 2015), and cannot be conclusively measured by single satellites. Because of these difficulties, our understanding of the growth of snow bedforms and their thermal effects must begin on the ground.

Systematic studies of snow bedform shapes and movement have, however, been separated by dozens of years and by thousands of miles. Cornish (1902) sketched snow bedforms, measured snow-wave wavelengths, and estimated the tensile strength of snow-mushrooms during a 3000-mile trip across British Columbia. Doumani (1967) photographed snow drifts, sastrugi, and barchanoids during two years' worth of traverses out of Byrd Station, Antarctica. Kobayashi (1980) used under-lit tables, photography, and meteorological stations to document the formation and advection of snow-ripples and snow-waves near their home institute of Sapporo, Japan. Finally, Filhol and Sturm (2015) collected photographs and descriptions of diverse snow bedforms, as well as LiDAR scans of dune fields near the University of Alaska, Fairbanks.

The past few years have seen a burst of new snow bedform research as several groups have deployed modern tools to document specific bedform behaviors. Kochanski et al. (2018) employed time-lapse imagery to detect bedform and sastrugi presence in the Colorado Front Range, and used this to predict the occurrence of snow bedforms as a function of weather. Amory et al. (2016) recorded the shift in sastrugi angle, and the resulting change in wind drag, during a storm in Adélie Land, Antarctica. Filhol et al. (2017) used a terrestrial LiDAR to capture fifteen images of an evolving field of snow dunes over the course of a 7.5 hour wind event in Finse, Norway. Hervé et al. (2014) and Naiim et al. (2017) noted that bedform evolution is rapid and difficult to capture with manual measurements, and have set up a terrestrial laser scanning system to automate the monitoring of sastrugi motion in the French Alps. These opportunistic observations have answered several basic questions about snow bedform evolution. For example, they have conclusively demonstrated that shifts in snow bedforms change surface wind drag (Amory et al., 2016), and that snow dunes merge and grow as they travel (Filhol et al., 2017).

None of the studies above, however, track the evolution of snow bedforms from one form to another. This makes it difficult to determine whether the observations represent rare events, or recurring patterns. Here, we aim to identify the overarching patterns of bedform evolution, so as to guide future monitoring studies. In particular, we seek to provide insight into the relationship between weather conditions and bedform evolution, so that, for example, future LiDAR studies can be deployed at the right times and places to capture the quantitative details of the most important events.

In this paper, we present the results of the longest-term observational study of snow bedforms that has been carried out to date: a three-year study on Niwot Ridge in the Colorado Front Range. Our observations are available at Kochanski (2018c). In this paper, we present detailed examples of the movement and formation of snow ripples, barchan dunes, snow-steps, snow-waves, and sastrugi. We analyze these observations in terms of the geomorphological processes that produce them (these processes are introduced in § S3). Finally, we outline a theoretical framework that describes the various evolutionary trajectories of snow bedforms in terms of the processes that drive their formation and growth.



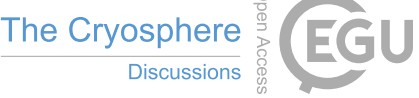

## 1.1 Names and natures of snow bedforms

The study of bedforms is still very new, and the names used to describe bedforms are occasionally controversial. Here we define the names that we use to organize our results. Example images are provided in § 3.1. We discuss only bedforms that we observed in Colorado.

*Plane beds* (§ 3.1.1), or flat snow surfaces, occur when snow falls during periods of light wind (Kochanski et al., 2018). These beds are smoother than the underlying surface, and may sparkle. This surface type is rare at our field site.

   *Snow ripples* (§ 3.1.3) are small transverse features (perpendicular to the mean wind) with wavelengths on the order of 15 cm. They are described in detail by Kobayashi (1980) and Kosugi et al. (1992). Filhol and Sturm (2015) refer to these features as 'ripple marks'.

*Barchan dunes* (§ 3.1.2) are crescent-shaped, or two-horned, dunes. They have well-defined crests, with a gentle upwind slope and a steep downwind slope that curves into two forward-pointing arms (Filhol and Sturm, 2015; Petrich et al., 2012; Doumani, 1967; Kobayashi, 1980; Goodwin, 1986). In this paper, 'barchan dune' (or 'isolated dune') refers specifically to crescent dunes that are separated from neighboring dunes by wide stretches of hardened snow or ice.

   *Close-packed dunes* (also § 3.1.2) have defined crests, but they are not separated by inter-dune patches of snow or ice. They 15 may be barchan dunes with arms melded into one another. We distinguish close-packed dunes from barchans because snow surface thermal properties (Petrich et al., 2012) and evolution are affected by exposed inter-dune snow.

   *Snow waves* (§ 3.1.6) are large transverse features with 5–15 m wavelengths (Filhol and Sturm, 2015). They propagate parallel to the wind (Cornish, 1902; Kobayashi, 1980), but extend for tens to hundreds of meters in a perpendicular or oblique direction. They may or may not have visible crests.

*Loose patches* (§ 3.1.7) is a catch-all term. We found that, when a surface was mostly hardened, any remaining loose snow frequently travelled in patches 1–5 m long. Unlike dunes and waves, these patches lack crests, and they tend to fill depressions in the existing snow surface rather than organizing into persistent structures.

   *Sastrugi* (§3.1.4) are the most widespread and well-developed erosional bedforms. Sastrugi are elongate features that present steep points into the wind (Filhol and Sturm, 2015; Amory et al., 2016). The points are regularly spaced, and the point of 25 each feature is aligned with the gap between two neighboring sastrugi immediately upstream. The word 'sastrugi' has been previously used to describe a wide range of snow features (Leonard, 2009), but its usage has become more focused in recent years.

   *Snow steps* (§3.1.5) are smaller, less regular erosional features (Kochanski et al., 2018). They present low (<2 cm) vertical faces to the wind, but lack the upwind-facing points that characterise sastrugi. Doumani (1967) called these features 'proto- 30 sastrugi,' but we avoid using this name because it implies a pattern of evolution, from steps to sastrugi, that we did not observe (see § 3.2).

   Finally, we introduce the *stealth dune* (§3.1.8). These are boomerang-shaped bedforms that resemble barchan dunes from afar, but their windward edges are vertical.



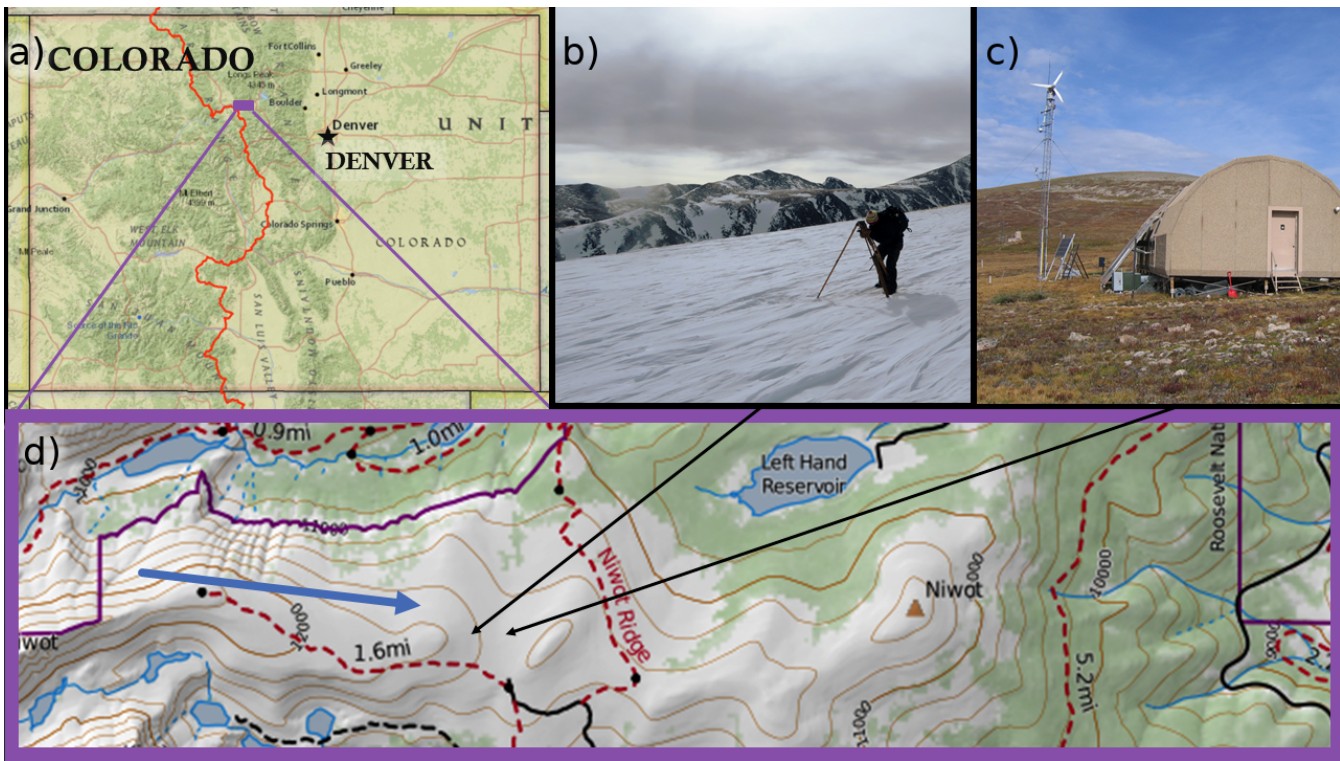

**Figure 1.** Field site on Niwot Ridge. a) Location of the field site within the state of Colorado. b) Installation of a time-lapse camera. c) Weather station. d) Close-up map of field site with arrows showing prevailing wind direction and locations of camera and weather station.

## 2  Methods

### 2.1  Field site

We surveyed bedforms on snow-crowned ridges and frozen reservoirs throughout the Front Range (§S2). Among the sites we visited, one stood out for its well-developed and undisturbed bedforms: the Niwot Ridge saddle site (Fig. 1, 40.054° N,

5  105.589° W). The site lies on the downwind end of a broad, treeless 3 km-long ridge, 5 km east of the Continental Divide, at an elevation of 3528 m. The ridge receives deep, dry snowfall that is shaped by consistent west-northwesterly winds. Snow falls from October through June, with the heaviest blizzards between January and March. Most blizzards bury the existing snow or ground surface. They deposit blank canvases on which bedforms evolve for a few days or weeks until they are buried by the next storm. The Colorado Front Range is a semi-arid region, with low humidity and famously dry and powdery snow. The wind

10  is primarily westerly, with local variations driven by temperature and altitude gradients over the continental divide. In winter the wind on Niwot Ridge blows almost without exception from the west-northwest. We draw all but one of the observations in the main body of the text from Niwot Ridge.





We also present one set of observations made on the frozen surface of Barker Reservoir, Colorado, where we observed a type of bedform that did not appear on Niwot Ridge (§ 3.1.8). Barker Reservoir sits at the bottom of an elevated (2483m) post-glacial valley and, in winter, it becomes a 1800 m-long stretch of ice. The valley funnels the wind west-southwest across the long axis of the lake. Much of the incoming snow is blocked by the town of Nederland, which occupies the upwind edge

of the reservoir.

## 2.2 Data collection

We observed bedforms on Niwot Ridge in person and through Day6 Plotwatcher Pro time-lapse cameras powered by cold-resistant Energizer Ultimate Lithium batteries. The cameras were active from February to March 2016, from October 2016 to March 2017, and from November 2017 to March 2018. They captured a total of 1082 hours of good visibility footage,

with photos taken every 10 s. Footage is missing at night, during white-outs, on days when clouds enveloped the ridges, when cameras were buried, and during equipment failures. Sample time-lapse videos are shown in video S1 and all videos are archived in (Kochanski, 2018b).

Weather and precipitation data are available from the Niwot Ridge Long Term Ecological Research Program station 200 m downwind of our cameras (Losleben, 2018a, b). Temperature measurements were collected with a Campbell Scientific CS500

mounted in a Stevenson Screen 1.5 m above ground, and wind speed was measured with an RM Young 501D mounted 7.5 m above ground. The recorded wind speeds from November through March average 10.5 m/s. The gusts average 14.4 m/s and reach as high as 29.7 m/s. The highest winds that we have seen sustained for an hour or more were 23.0 m/s. Winter temperatures average -7.5°C and vary from -28–11°C. Melting temperatures occur in every month.

We measured bedform sizes and velocities both in-person, using meter sticks, and by reference to poles with 10-cm stripes

placed in front of the cameras. We used Tracker video analysis software (Brown, 2018) to extract high-resolution position-velocity data from several of our images. In one analysis we used the video to make a frame-by-frame visual estimate of the variations in the flux of blowing snow. We only attempted these analyses when the features moved perpendicular to the camera view, to avoid distortion by distance. We have endeavored to convey all sizes, speeds, and intensities in the most useful units that are available to us. The quality of these measurements varies, unfortunately, between data sets; for example, in many

of our images the measurement poles have blown away. We analyzed the resulting data with Matplotlib (Hunter, 2007) and NumPy (Oliphant, 2007). We previously presented a statistical analysis of the relationship between snow bedforms and weather conditions on Niwot Ridge in Kochanski et al. (2018). In this paper we focus on bedform dynamics.

## 3 Results

### 3.1 Bedform movement

Here, we illustrate the movement of each bedform with examples from time-lapse footage: planar snow surfaces (§ 3.1.1) barchan dunes and close-packed dunes (§ 3.1.2), ripples (§ 3.1.3), snow-waves (§ 3.1.6), sastrugi (§ 3.1.4), snow-steps (§ 3.1.5),





and stealth dunes (§ 3.1.7). For each example, we include still frames in the text. These still frames have been selected for clarity, avoiding mist and blowing snow, and have been edited for color and contrast. The frames are taken from the videos provided in 5 minute supplementary video S1 (Kochanski, 2018a). As this paper focuses on the motion of snow bedforms, video S1 is the most important figure.

### 3.1.1 Plane beds

The falling snow that we observed often settled into plane beds. These snow surfaces were smoother than the ground, or older snow, that they covered, and were unmarked by corners, edges or organized bedforms. They are also the only surface type made from unsintered, unbroken snowflakes, and thus the only surfaces that sparkle in the sun. These surfaces did not persist once any perceptible amount of snow began to blow.

### 3.1.2 Snow dunes

The biggest dune that we saw was 54 cm tall (Fig. 2a, in a field of 30–50 cm dunes). The smallest dune (Fig. 2c) was no more than 7 cm tall and 40 cm long.

Fig. 2b shows a field of close-packed dunes. These advected downwind, but they also interacted with one another. They rapidly merged, calved, and changed height. We tracked the heights, widths (measured horizontally from the dune crest to the downwind foot of the dune), and velocities (measured at the crest) of the dunes in Fig. 2b; the results are shown in Fig. 3. The dunes were roughly self-similar. Their velocities, unlike the velocities of sand dunes (see § 4.2) were not correlated with their sizes.

Fig. 2d shows the motion of of a field of barchan snow dunes. These dunes are not close packed, and do not interact with one another. They maintain their shapes as they move downwind (see video S1, 1:07).

We tracked the position of the barchan dunes in Fig. 4 through time. The tracked points on the dunes moved an average of 0.63 m in the first fifteen minutes of observation, for an average velocity of 2.52 m/hr, after which they decelerated until sunset.

Neither the weather that drove the dunes nor their speeds were constant. The dunes were subject to frequent gusts. The instantaneous velocities of the dunes, observed over 10s intervals, varied from 0 to 15 m/hr. The dune arms (A1 and A2) accelerated and decelerated at the same times as the peaks (C1 and C2), which indicates that the dunes moved as a whole in response to changes in the wind. Although our wind data measurements were not of sufficient resolution to resolve these gusts, the gusts were associated with pulses of blowing snow. We evaluated the blowing snow flux visually in our camera footage at 10 s intervals, and indicate them with gray shading in Fig. 4b. The shading varies from white (no visible snow—though in our experience, the flux of blowing snow can be quite intense before it is opaque enough to appear on camera) to dark gray (whiteout). We cross-correlated the intensity of blowing snow (on a normalized scale from 0 (white, no blowing snow) to 1 (dark grey, whiteout)) with the instantaneous velocities of the dunes (Fig. 4c). This revealed that the velocities of C1, C2, and A2 lagged the pulses of blowing snow by $30 \pm 5$ s, though the velocity of A1 did not correlate with the blowing snow. This positive correlation and positive lag is evidence that high fluxes of blowing slow cause high dune velocities, instead of the other way around.



**Figure 2.** Snow dunes on Niwot Ridge, CO. a) 30–40 cm tall stationary barchan dunes on 03/03/2018. Temperatures are near freezing. The snow between the dunes, which has been marked by erosional bedforms, is now unmarked by the author's weight. b) Time-lapse of dunes travelling downwind on 22/01/2018 from 13:29–13:34 (video S1, 0:08). No hard or eroded snow is visible between the dunes. c) Smallest barchan dune observed on Niwot Ridge, seen moving downwind at 11:25 09/11/2017 at 1–2 cm/min. <1–5 mm wide snow grains creep across the crest (video S1, 0:57). d) Time-lapse of barchan dunes from 07:20–08:20 on 17/01/2017 (video S1, 0:39). Lines track the positions of the dune crests.





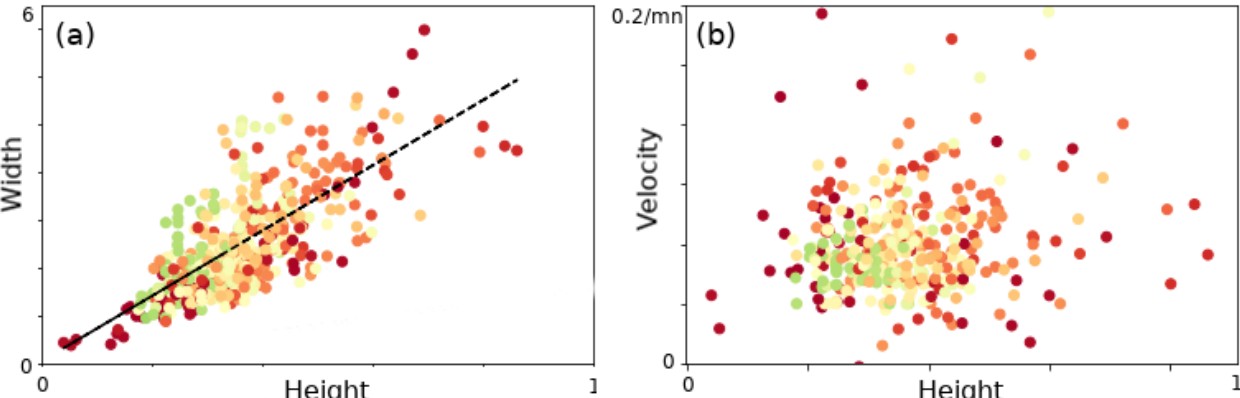

**Figure 3.** Scaling relationships for dunes observed on 22/10/2018. Length is measured in an arbitrary unit of approximately 30 cm. Colors indicate the distance of the dune from the camera (dark red is close, pale green is far) which should be independent of the measured quantities.

The snow on the moving dunes that we observed demonstrated all three major types of aeolian snow movement. Saltating snow is clearly visible in the time-lapse footage used for Fig. 2b. Snow grains crept, with halting movements, across the crest of the dune in Fig. 2c. Finally, the images used to make Fig. 2a were often obscured by blowing snow that was suspended to at least the height of the 2 m camera. Although blowing snow is not always visible on the cameras, in person we observed moving snow dunes, ripples, and waves only during ground blizzards, and never on days without blowing snow.

### 3.1.3 Snow ripples

Ripples appeared at our study site both as a primary features, covering wide swaths of ground (Fig. 5c), and as a secondary feature that adorned dunes and snow waves (§ 3.1.3). The ripples that we observed were 0.5–2 cm tall with wavelengths of 10–25 cm.

Fig. 5a shows ripples covering the snowy portion of partially snow-covered ground in the early part of the snow season. The ripples advected downwind. Dislocations propagated across the bed, leading individual ripples to merge and separate from one another. The velocity of the ripples was consistent throughout the observation; although scale bars are missing from this image, the crests advanced with a frequency of $1.81 \pm 0.16$ min$^{-1}$. Despite this advection, the snow did not advance noticeably over the rocky part of the ground, implying that the mass transported by the ripples was either buried or lost to suspension at the edge of the rocks.

The photos in Fig. 5b were also taken on a day when both snow and bare ground were exposed on the ridge. They formed from a mix of wind-blown snow and sand. The sand (dark) was concentrated on the crests of some of the ripples. Over the course of this 3-minute observation, the ripples without sandy crests faded away, leaving mostly sand-covered ripples. Both sand-covered and sand-free ripples travelled at $4 \pm 1$ m/hr.



**Figure 4.** Movement of two barchan dunes on Niwot Ridge, Colorado, from 16:03 to 16:28 19/01/2018, and analysis of the effect of pulses of blowing snow on dune movement. See video S1 1:07. a) Dunes at 16:03. Colored lines show the crest position through time, taken from time-lapse imagery. Poles have 10cm stripes. Horizontal pole is held 2m in front of vertical poles, which are 2m apart. b) Horizontal (left to right) positions of the dune crests at points C1, A1, C2 and A2 at 10s intervals. Background is shaded when blowing snow is visible in the footage. c) Cross-correlation of the presence of blowing snow with the instantaneous velocities of the dunes.





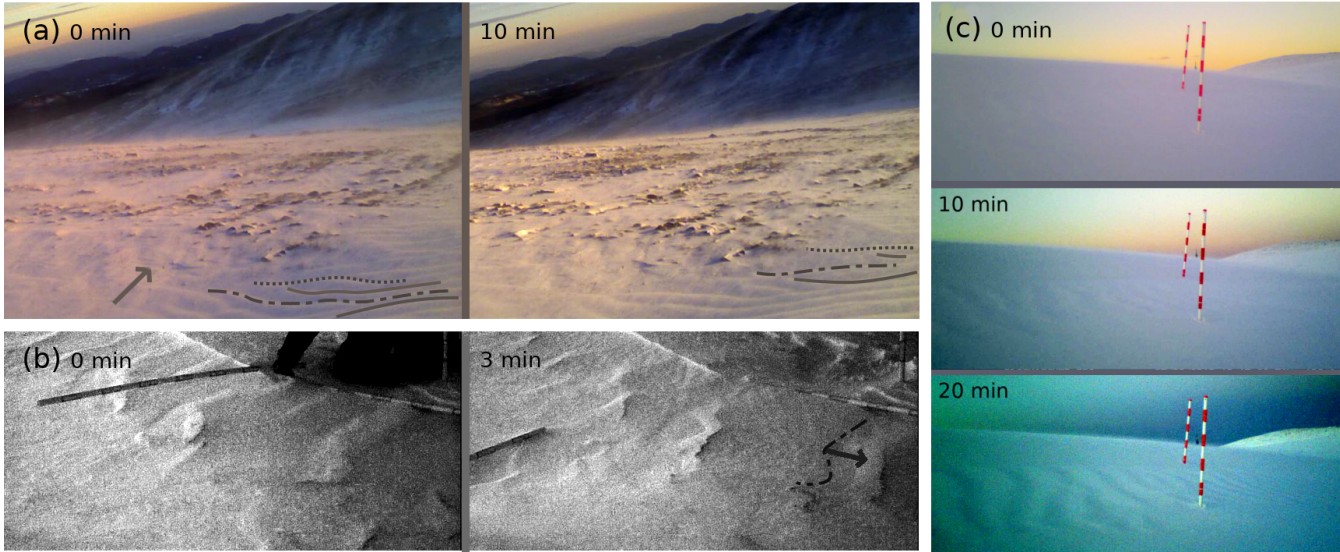

**Figure 5.** Time-lapse images of snow ripples on Niwot Ridge, Colorado. (a) Ripples advect from 7:28—7:38 on 19/11/2016, video S1 1:32 (b) Sand covers the crests of advecting ripples from 17:23:33–17:26:33 on 18/01/2018, video S1 2:06. Pole has 10 cm stripes. (c) Ripples emerge from a previously flat snow-covered surface on Niwot Ridge between 18:05 and 18:25 on 27/03/2018, video S1 2:23. Poles are 2 m apart and have 10 cm stripes. Ripples travel many wavelengths between photos.

Fig. 5c shows snow ripples emerge from a flat snow surface. They advected downwind by many wavelengths between photos, and their amplitudes grew gradually over time.

### 3.1.4 Sastrugi

Our study site, Niwot Ridge, frequently held sastrugi that were 14–40 cm deep and spanned 45–90 cm between points. The largest observed sastrugi (at the downwind end of Niwot Ridge) were 90–120 cm deep, and spanned 120–180 cm. The troughs between those sastrugi resembled meandering slot canyons.

Fig. 6 shows two erosion events in which sastrugi retreated downwind. We tracked the shapes of the twenty best-defined points in Fig. 6a for two hours, and found that five became more overhung, seven became less overhung, and eight retreated without changes to their profile. The majority of the sastrugi in Fig. 6b maintain straight vertical edges, but two became more gently sloped and one became steeper during the 30 minute observation.

When we observed sastrugi in person, during high-wind events, we saw loose snow collect at the bottoms of their points. These loose snow grains moved continuosly, such that the loose snow sparkled with motion while the sastrugi themselves appeared dull. Up close, we saw that large grains collected at the base of the sastrugi, then crept downwind along both sides of the point. This creep moved the grains in an oblique downwind direction, without moving them vertically over the sastrugi. These patches of loose snow disappeared momentarily in strong gusts.





**Figure 6.** Time-lapse of sastrugi crests (a) from 15:30 (pictured) to 17:30 on 18/11/2016, video S1 2:51, and (b) from 10:23 to 10:53 (pictured) on 25/03/2016, video S1 3:05.



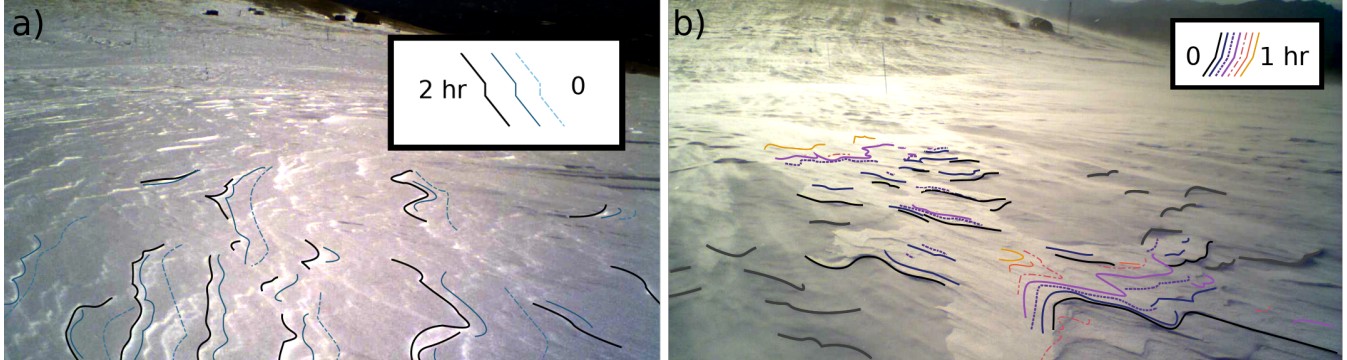

**Figure 7.** The movement of snow-steps on Niwot Ridge, Colorado (a) 21/03/2016 12:06–14:06 (video S1 3:35, photo at end of observation) and (b) 13/03/2016 07:09–08:09 (video S1 3:57, photo at beginning of observation).

We did not observe sastrugi formation. Our video observations therefore provide evidence that sastrugi are stable enough to translate by several times their height without significant changes of form, but they do not show direct evidence of the mechanisms of sastrugi formation.

### 3.1.5 Snow-steps

Fig. 7a shows a row of slowly moving snow-steps left in the wake of a traveling snow-wave (snow-step formation behind waves is discussed in § 3.1.6). The steps retreat downwind over the observation period. When they first form, at the beginning of the observation, they are relatively straight and soft-looking. Over the following two hours, they become more crenulated and sinuous in plan view. The density of the snow-steps on the surface also increases; Fig. 7a shows all of the steps visible at the start of the observation (dashed blue lines), which is not all of the steps visible at the end (photo, select steps traced in

black).

Fig. 7b shows differential step motion in a case where an old, hardened snow layer (duller white) lies under a relatively new and soft layer (brighter white). The newer steps eroded during the day, but the older steps did not move visibly. At the end of the day the newer steps were entirely removed. Their removal revealed a subtle landscape of older snow-steps that had apparently been buried and persisted under the snow.

### 3.1.6 Snow-waves

Snow-waves appeared frequently on Niwot Ridge. The waves we observed were 1–3 m long parallel to the wind , with crests separated at wavelengths of 10–20 m. Fig. 8a shows a field of snow-waves extending for tens of meters in a direction oblique or perpendicular to the wind.

Fig. 8d shows a section of a wave. The wave has several separated crests, joined by low, rippled sections of the wave. The

crests point directly downwind, but the overall orientation of the wave is oblique.



**Figure 8.** Snow-waves on Niwot Ridge, CO. (a) long-distance view on 9/11/2017 with estimated wind direction and scale (b) time-lapse image showing motion of wave (shaded) and small erosional features (lines) away from the camera from 12:20 (blue, dashed) to 13:00 (purple, solid) on 24/02/2017, video S1 4:19 (c) parallel snow-steps in the wake of a wave on 9/11/2017 (d) alternating wave crests and ripples on 12/11/2017.





Snow waves move by advecting downwind, but they also interact with the snow surface beneath them. We document this interaction in Figs. 8b, c and 9. Fig. 8b shows time-lapse imagery of the movement of a snow-wave, and nearby snow-steps. As the video progressed, the visible steps retreated, and new steps appeared behind the wave.

Fig. 8c shows this process close-up. The travelling wave in this photo buried the snow-steps to its left. It also genererated the snow-steps to its right. The right-hand steps parallel the wave crest, and the steps further from the crest are slightly more crenulated than the steps at the top. The snow surface behind the wave was several centimeters higher than the snow in front, implying that the new steps formed in newly-deposited snow.

Fig. 9 further illustrates the interaction between snow-waves and the underlying snow surface. The wave started as a low rippled section. The ripples occasionally appeared, disappeared, and merged, which was consistent with a side view of ripple dislocation (e.g. Fig. 5a). The ripples travelled about ten times as fast as the bulk of the wave. After about 60 minutes, the last ripple caught up to and merged with the previous ripples, forming a single crest. The velocity of the wave did not appear to change as it shifted from a rippled section to a crest. The crest deposited a layer of new snow, which was noticeably higher and fresher-looking than the previous snow surface. Several snow steps formed in this freshly deposited snow. They retreated downwind and decelerated over time. Fig. 9b tracks the peaks of the ripples, crest, and snow-steps at 5–10 minute intervals.

### 3.1.7 Loose snow patches

Small quantities of loose snow on deeply eroded surfaces collected into longitudinal "patches" a few centimetres thick. They flowed between underlying sastrugi and do not have persistent forms. The patches that we observed were 25–300 cm wide and 0.5–10 m long. On one occasion, a set of otherwise stationary snow-steps began to retreat when a patch of snow passed by.

### 3.1.8 Stealth dunes

Finally, we observed an extreme erosional bedform that we propose to call the *stealth dune* for its low profile and its rarity (Fig. 10). These dart-shaped dunes sit on ice and resemble barchans from afar, but their slopes are inverted: the upwind edges of the dunes are hard and vertical, while their lee sides are nearly flush with the ice. Figs. 10a–c show three dunes from a field of mixed size. The wing spans of the stealth dunes in that field ranged from 0.15–3 m, and their heights ranged from imperceptible thinness (likely due to scouring by wind after the dune formed) to a maximum of 10 cm. Fig. 10d shows an idealized stealth dune, emphasizing the vertical upwind slope. We observed stealth dunes only on the frozen mile-long surface of Barker Reservoir, and we did not see them move.

### 3.1.9 Bedforms in spring

We observed snow bedforms from first snowfall through late March. When week-long periods passed without snowfall, suncups appeared on the snow surface. These surfaces were consistently covered by sastrugi. The suncups appeared to be symmetric around the sastrugi, including the underhung sides of lanceolate sastrugi: they were not visibly larger on the sunny south sides, nor eroded on the upwind sides.





**Figure 9.** A rippled wave travels across Niwot Ridge from 6:40–8:40 on 01/19/2017, video S1 4:40. The wave is covered by ripples, which eventually collapse into a single crest, and deposits fresh snow as it passes. This in turn develops snow-steps.

From April or May through July, the surface of the snow on Niwot Ridge deflates due to melting and sublimation. We are missing observations in April and May, but in June 2016 and 2017 the surface of the snow was soft, wet, and marked by suncups without other bedforms.

The rate of sublimation on the ridge is slow compared to bedform formation. We observed the snowpack on Niwot Ridge shrink in above-freezing temperatures from 19/06/2017–01/07/2017. The snow depth decreased by 1.1 m during this 12-day period (0.092 m/day). As many bedforms that we observed grew in periods of hours, it seems unreasonable that they are erased by sublimation in spring. We therefore infer that spring conditions prevent bedforms from forming in the first place. This agrees with other authors' observations that bedforms grow best in dry snow.



**Figure 10.** Stealth dunes viewed from several angles on Barker Reservoir, Colorado, on 16/01/2016. (a) Large stealth dune (b) Remnant of small stealth dune (c) Marks left on ice beneath a dune; surface is raised by 1.5 cm (d) Schematic of stealth dune.



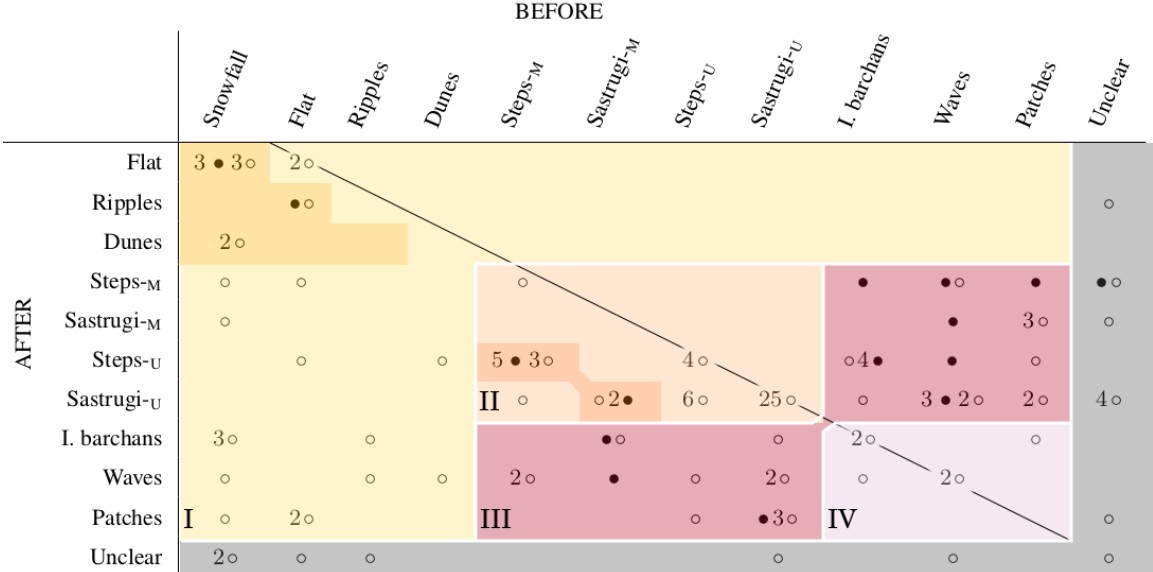

**Figure 11.** Transitions from one type of bedform to another, either observed directly (●) or implied during a <12 hr gap in footage (○, e.g. overnight. Snow-steps and sastrugi may be moving (M) or un-moving (U) when observed. Colored regions I-IV group similar transitions. 'Dunes' are close-packed (as in Fig. 2b), 'I barchans' are separated (as in Fig. 2d).

## 3.2 Bedform evolution

In § 3.1 we showed three examples of transitions from one type of bedform to another. In § 3.1.1 we discussed the deposition of flat snow surfaces during heavy snowfall events with gentle winds. In Fig. 5c, we showed ripples emerging from a flat snow surface. Finally, Figs. 8b–d and 9 showed snow waves burying existing snow steps and leaving new snow steps in their wakes.

The sum of all the transitions that we observed are presented in Fig. 11. These include transitions that we observed directly (●), as in the example cases above, plus changes in the surface that happened overnight or during other <12 hr gaps in our observations (○).

Some of the observed transitions appear to be irreversible. For example, we saw flat surfaces turn into ripples (Fig. 5c), but did not see rippled surfaces return to flat. We also saw moving snow-steps and sastrugi stop moving (region II) but did not see

them start moving again. Moreover, at least some of the snow-steps that we observed above decelerated continuously (Fig. 9b) with no obvious outside influence, implying that they stop spontaneously.

The indirect transitions (○) imply that ripples, close-packed dunes, and flat surfaces must turn into various combinations of snow-waves, barchan dunes, loose patches, snow steps, and sastrugi, and that snow dunes must somehow give way to snow-waves and vice versa. We did not observe any of these transitions directly. Moreover, we did not observe the formation of

close-packed dunes, although we did once observe an indirect transition in which they appeared shortly after snowfall and a brief camera whiteout.



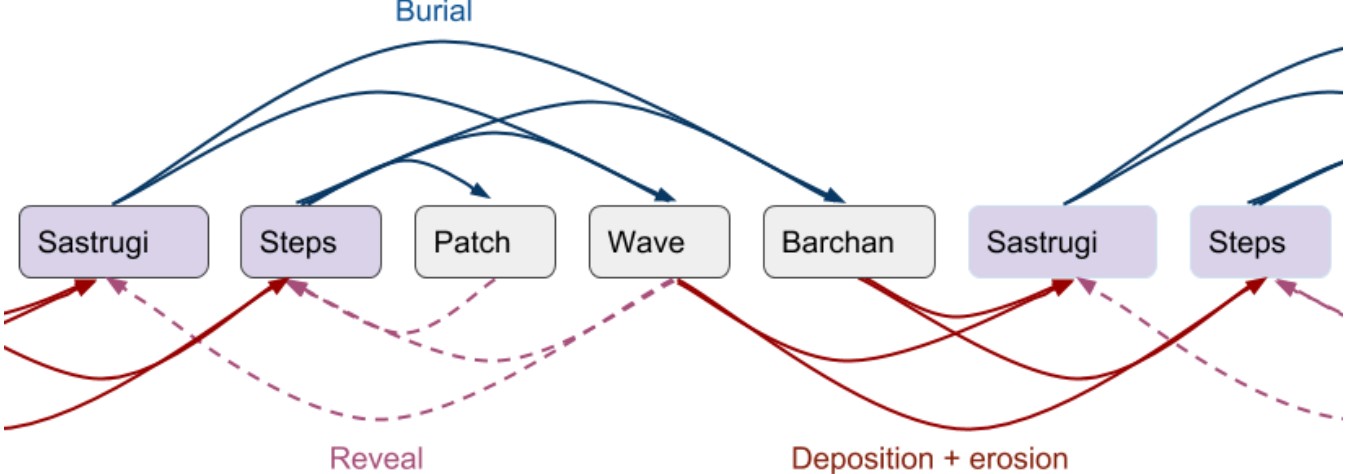

**Figure 12.** Observed interactions between erosional bedforms and bedforms made of loose snow. Left-to-right interactions result in the net deposition of new snow on the surface. Burial (blue): a wave, dune, or patch advects onto an eroded surface and covers the existing bedforms. Reveal (dashed): the wave, dune, or patch continues traveling, and leaves the observed area without modifying the surface. Deposition + erosion (red): a wave, dune or patch deposits a layer of freshly-accreted snow onto the surface. Erosional bedforms are carved immediately in the fresh snow.

These results demonstrate that bedform evolution is cyclic. The diagonal line in Fig. 11 represents 1:1 transitions. If bedform evolution were unidirectional, we would be able to arrange the table such that all observed transitions, both direct (•) and indirect (○), lie below the diagonal line. Our observations, however, cannot be organized so tidily. The cyclic behavior is driven by the transitions in Region III (shown graphically in Fig. 12). We saw snow-waves, snow patches, and barchan dunes bury

snow steps (such as in Fig. 8c) and sastrugi. Sometimes, the snow-waves or loose patches moved on without changing the surface, and revealed the buried bedforms, apparently unchanged. On one occasion, existing snow steps were re-mobilized by a passing loose patch. Barchan snow dunes always deposited a new snow layer in their wakes, and snow waves sometimes did. These new layers eroded into either sastrugi or snow-steps.

## 4  Discussion and directions for future work

Many snow bedforms are analogous to other self-organized aeolian features. Barchan snow dunes, close-packed dunes, and snow ripples resemble sand dunes and ripples. Snow-steps and sastrugi find their analogues in scoured bedrock: the scalloped edges of snow-steps loosely resemble bedrock fluting, and the aerodynamic points of sastrugi resemble yardangs. Other bedforms, such as snow-waves, do not have obvious analogues elsewhere in the aeolian world, but mediate the transitions between the sand-like and bedrock-like bedforms.

We use these observations to place snow bedforms into three categories that we use in the remainder of this discussion:





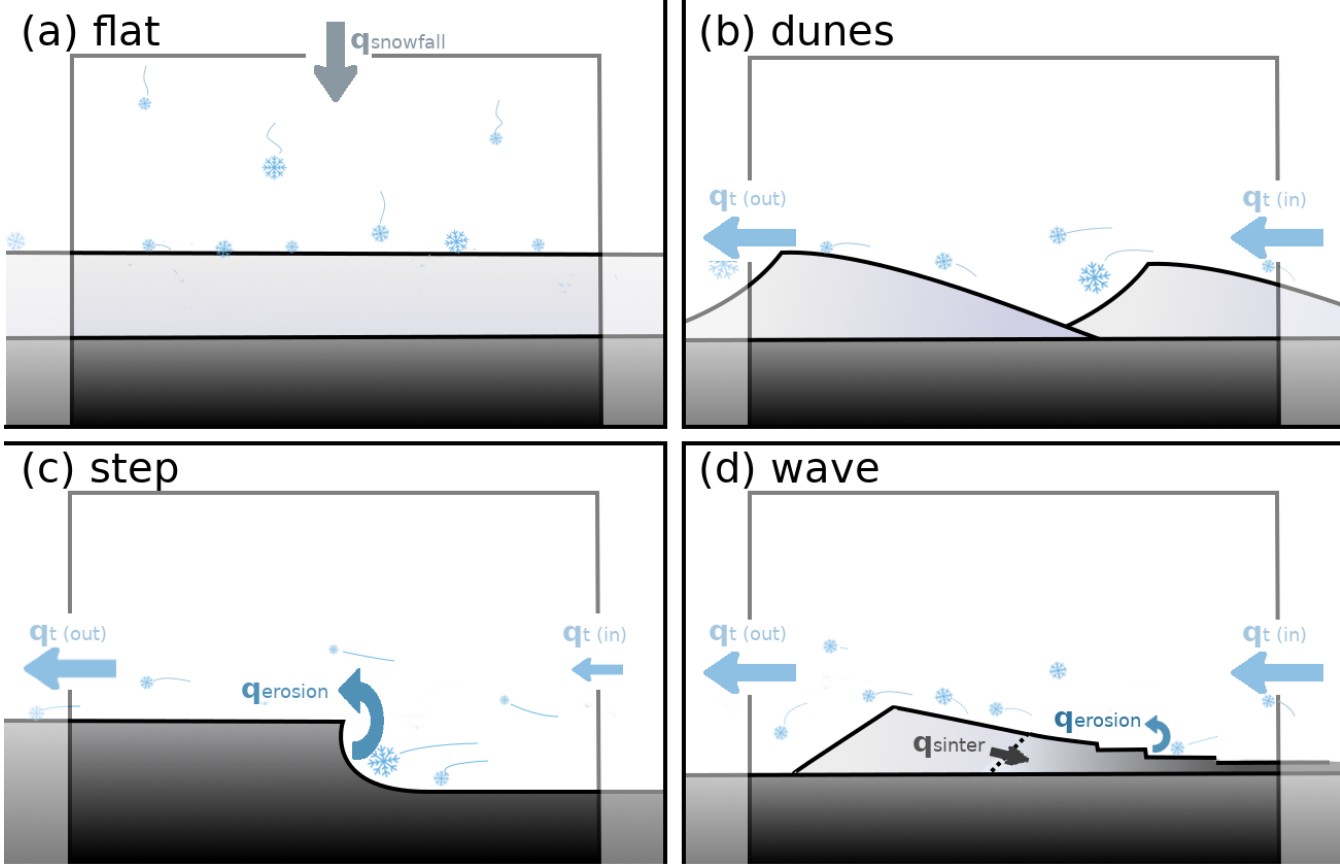

**Figure 13.** Expected self-organization of snow in environments dominated by (a) snowfall, (b) aeolian transport, (c) erosion, or transport on a cohesive surface, and (d) a mix of processes. Here, $q$ refers to the total flux of snow, and $q_t$ refers to aeolian transport by creep or saltation.

- *Loose-surface bedforms* consist entirely of granular snow, which creeps or saltates downwind and slips when over-steepened. The bedforms on these surfaces resemble aeolian sand features. These include close-packed dunes and ripples.

- *Hardened-surface bedforms* are cohesive, solid, and shaped by erosion. These include snow-steps and sastrugi.

- *Mixed-surface bedforms* consist of loose snow traveling over a hardened-snow surface. Loose-surface and hardened-surface bedforms alternate in patches or stripes. These include isolated barchan dunes and snow-waves.

Fig. 13 illustrates the major snow fluxes that shape a representative bedforms from each category. Bedforms are shaped by at least four major processes: snowfall, aeolian transport, erosion, and sintering. When snowfall dominates (Fig. 13a, § 4.1), snow settles flat. When snow is granular and blown by the wind, it forms loose, sand-like bedforms (Fig. 13b, § 4.2). When snow is hardened, the surface evolves by erosion (Fig. 13c, § 4.3). When snow rests in place (not pictured) it sinters and becomes more cohesive (Blackford, 2007). Finally, many snow bedforms are shaped by more than one of these processes (Fig. 13d, § 4.6).



The horizontal flux of snow is distributed between bedload and suspended load. Suspension rates depend on the shear velocity of the wind, $u_*$, which is a function of the wind speed and the surface roughness length $z_0$. Assuming a logarithmic wind profile, the wind speed $u$ measured at height $z$ is $u = \frac{u_*}{\kappa} \ln(z/z_0)$, where $\kappa = 0.4$ is the Von Kármán constant. We assume the roughness length $z_0$ is 0.2 mm; this is a typical value for wind blowing perfectly parallel to a sastrugi field (Inoue, 1989b) or

for freshly fallen snow with small drift features (0.24 mm, Gromke et al. (2011)), but much higher than the standard values for planar snow ($\sim 0.5$ mm), and much lower than the values for wind blowing perpendicular to sastrugi (1 mm, Inoue (1989b)). For our field site, this gives us a shear velocity of $u_* = 0.4$ m/s in the average 10.5 m/s measured wind, and $u_* = 0.9$ m/s in a high 23 m/s measured wind.

Particles begin to be suspended when the shear velocity is greater than their settling velocity, and they are transported fully in

suspension when the shear velocity is twice their settling velocity (these thresholds are equivalent to Rouse number thresholds of 2.5 and 1.25). The settling velocity of fresh snow is 1–2 m/s (Barthazy and Schefold, 2006), although saltating snow tends to break into ice fragments (Comola et al., 2017); assuming these fragments are spherical and move much like water droplets, their settling velocities increase non-linearly with their size, and range from 0.1 m/s (0.05 mm diameter) to 0.3 m/s (0.1 mm) to 1.1 m/s (0.5 mm) (Anderson, 2008). We therefore expect that large snow particles ($\geq 0.1$ mm) are consistently transported

as bedload, and that particles of 0.1 mm diameter and smaller are often suspended.

### 4.1 The rarity of flat snow surfaces

Flat snow surfaces are rare on Niwot Ridge, and occur only when flat snow falls during periods of gentle winds. We never saw snow blow across a flat snow surface, and suspect that any wind that lifts snow grains from the surface also initiates bedforms in the snow. We therefore hypothesize that flat snow surfaces occur only when the shear velocity of the wind is below the

entrainment threshold for the snow. Li and Pomeroy (1997) estimated that for dry snow this threshold is equivalent to a 7.7 m/s wind speed, measured 10 m above the ground. Kochanski et al. (2018) found that flat snow on Niwot Ridge occurs at wind speeds at or below 7.3 m/s measured at 7.5 m above the ground, though the threshold is time-dependent.

### 4.2 Snow dune dynamics

Loose-snow bedforms bear many similarities to sand dunes and ripples. Here, we focus on snow dunes; for a thorough and

quantitative analysis of snow ripples, see § 3.3 of Kobayashi (1980). Our observations show that the sand dune analogy provides at least a moderately useful description of snow dune physics. Many of the processes that shape sand dunes appeared around the snow dunes that we observed, including saltation, suspension, creep (§ 3.1.2), and grain sorting (Fig. 5b). In the following paragraphs we discuss three major differences between sand dunes and snow dunes, and present conceptual models to explain these differences.

First, snow dunes are small. The dunes we observed varied from approximately 7–55 cm in height (see examples in Fig. 2). We saw at least one well-formed dune barely longer than my snow-shoes (about 40 cm, see 2c). Other reports from Antarctica have also documented snow dunes only tens of centimeters high (Doumani, 1967), even in snowy, cold environments where dunes should have considerable time to grow. Some but not all of the size difference may be attributed to known scaling laws.



The fundamental length scale for sand saltation and sand dune growth is $\lambda_{\max} = 50 d \rho_s / \rho_f$, where $d$ is the grain size, $\rho_f$ is the fluid air density, and $\rho_s$ is the solid grain density (Elbelrhiti et al., 2005). Saltating snow grains on Niwot Ridge have an effective density of perhaps half the density of ice (450 kg/m$^3$), occur at low air density due to altitude (0.860 kg/m$^3$ at -5$^o$C), and are perhaps 0.2 mm in diameter, leading to a value of $\lambda_{\max} \approx 5.2$ m. For contrast, a typical value of $\lambda_{\max}$ for Sahara sand

is 20.4 m (Elbelrhiti et al., 2005). The snow dunes that we observed, however, were not even a quarter the size of Sahara sand dunes. We therefore infer that at least one process that is present in snow but not in sand limits the size of dunes.

Second, sand dunes have downwind slip, or avalanche, faces, but not all of the snow dunes that we saw were steep enough to slip. The dunes in Figs. 2a and c clearly have steep slip faces, but the dunes in Fig. 2d appeared to be nearly flush with the snow. The dunes in Fig. 2b and Fig. 4 are indeterminate.

Third, the speeds of sand dunes are inversely proportional to their heights (Bagnold, 1937; Vermeesch and Drake, 2017). We do not observe this relationship in snow dunes. Although Fig. 9 clearly snows small ripples moving faster than the larger bedform that they cover, when we explicitly tracked the velocities and heights of a field of dunes (Fig. 3), we did not find a systematic dispersion relationship.

These three features — dispersion, size, and avalanching — are all manifestations of the pattern of mass flow around a dune.

The dispersion relationship for sand dunes is a direct function of the conservation of sand flux (Bagnold, 1941; Vermeesch and Drake, 2017). If all dunes in a field are exposed to the same flux of blowing sand, and all of them trap this same flux on their lee slopes, then their velocity will be inversely proportional to their height. Thus, as we document that snow dune velocities are not inversely related to the dune heights, we infer that snow flux is not conserved within individual snow dunes. Lack of snow conservation within a dune might occur when (1) a dune loses mass to sublimation (2) gains mass from snowfall, (3) the

driving wind speed varies rapidly, or (4) a dune exchanges mass with its neighbors.

Our observations indicate that sintering is the primary limit on dune size on Niwot Ridge, although we cannot discount the removal of particles into suspension. Snow dune size could plausibly be limited by three processes: sublimation, suspension and sintering. We have considerable evidence that snow sinters in dunes. We saw several hardened barchan dunes; the dunes in Fig. 2a were hard enough to support the author's weight. We also saw hints of a snow stratigraphy and cohesive snow-steps

in the wakes of the mobile dunes in Fig. 2c and Fig.2d. In contrast, we did not find evidence that sublimation limits the size of snow dunes. The sublimation rates on Niwot Ridge are very slow relative to the speed of snow dunes (§ 3.1.9), and we know that dunes do not grow larger in very cold locations, such as Antarctica, where sublimation rates are yet lower than on Niwot Ridge. As for suspension rates, it seems likely that the fraction of loose snow in suspension increases over time as snow particles fragment and shrink. The Rouse number calculations formed at the start of § 4 indicate that grains $\geq 0.1$ mm are

consistently transported in bedload; in our experience, grans of this size and larger are very common in dunes and bedload, but without quantitative measurements of grain size we cannot constrain the rate of change of typical grain sizes through time.

We hypothesize that low-lying, non-avalanching, non-flux-conserving dunes could be formed by rapid changes in the wind speed. In order for a dune to have a downwind avalanche face, many wind-blown grains must fall out of the air onto the avalanche face. In sand dunes, grains fall out of the air in the recirculation zone downwind of the dune (as in Fig. 14a). The

recirculation zone in turn is generated by the strong negative step in the bed formed at the brink of the dune. If the wind rises,




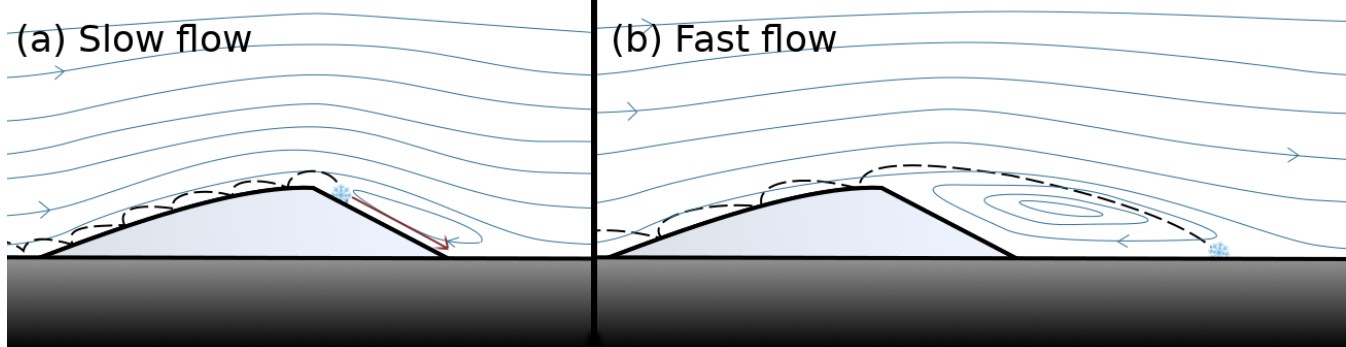

**Figure 14.** Conceptual model of grain transport over a dune in (a) slow-moving air and (b) fast-moving air. Solid blue lines are time-averaged streamlines, with boundary-layer fluctuations omitted for clarity. Dashed black line shows a grain trajectory. The arrow in (a) shows an avalanche path.

however, the recirculation zone will lengthen, and the hop lengths of saltating grains will lengthen, while the length of the slip face stays the same. Saltating grains may then miss the slip face entirely and collect in a drift downwind of the dune (Fig. 14b). This is only possible in small dunes whose slip faces are similar in length to the saltation hop length. If too large a number of grains miss the slip face, the dune will lose mass and dwindle. Therefore, small, non-avalanching dunes are likely transient

features. The final shape of a snow surface, however, is usually set within one or two days of snowfall (Filhol and Sturm, 2015), so even short-lived features may play important roles in the surface evolution.

## 4.3 Snow-step erosion

All of the hardened-snow bedforms that we have observed are characterized by steep windward edges. We hypothesize that these steep edges hold the key to a general understanding of snow surface erosion. Our observations have shown that snow-

steps and sastrugi move by retreating directly downwind, which indicates that erosion is concentrated on the vertical steps. In Fig. 15, we present a conceptual model that demonstrates how this erosional pattern could be generated around a snow-step.

The flow pattern around our conceptual step is modelled after the flow around an upwind-facing step in steady laminar flow. Such steps create two flow detachments: one on the upwind side of the step, where flow stagnates, and one on the top of the step. Each flow detachment is associated with a recirculation zone (shaded in Fig. 15a) with slow, overturning flow. We have

ommitted boundary-layer eddies and turbulence for clarity, and have neglected the loss of flow momentum to saltating particles and the porous snow.

This flow pattern offers several opportunities for particles to detach from the flow and strike the area around the step. When the flow encounters the step, it reaches an equilibrium state in which the pressure in the stagnation zone is just high enough to deflect the flow past the step. Snow grains, however, are 200–1000 times denser than air, and thus will be accelerated 200-

1000 times more slowly by a given force; the pressure gradient that lifts the air flow will barely deflect the grains. Thus, if the incoming angle of a particle is sufficiently low, that particle strikes the step. This process has been modelled in detail for




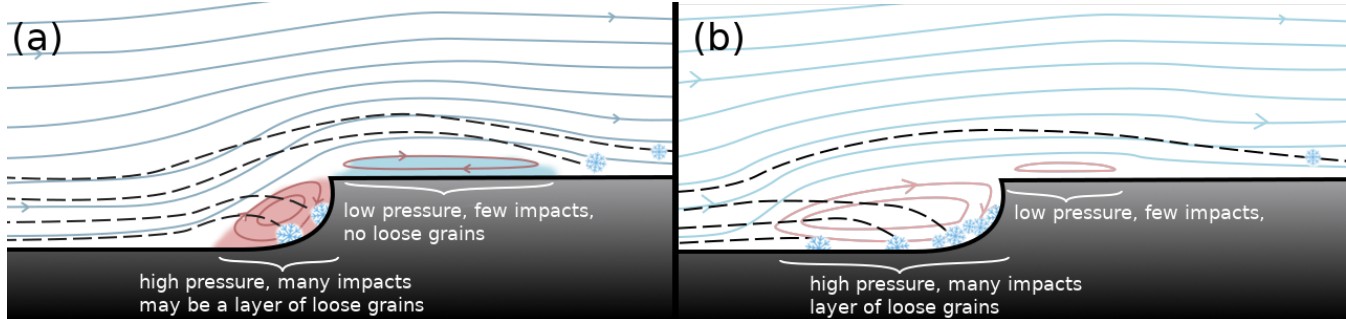

**Figure 15.** Conceptual model for air and particle flow over a snow-step in (a) high wind and (b) low wind. Solid lines show idealized streamlines over the step, including two regions of detached flow (shaded in a). Dashed black lines show snowflake trajectories.

flow around cylinders, spheres, and disks (Langmuir and Blodgett, 1946; May and Clifford, 1967). Around the forward-facing step, we expect that the highest impact rates will occur before the step and on its vertical face, where gravity assists inertia in detaching particles from the flow. We expect that the lowest impact rates will occur on the top of the step, as particles that successfully cross the step must have upward momentum that will delay their fall.

Only the most energetic of the detached particles will strike the vertical face of the step. Particles that detach from the flow lose energy to air resistance. Any particle that loses too much energy will fall out of the flow without striking the step. These particles are unlikely to carry enough energy to cause erosion. The length of the stagnation zone, and therefore the energy it removes from detached particles, increases with the height of the step and decreases with the wind speed. We therefore expect particles to reach the step at more-than-linearly greater rates in high winds, when the particles have high initial energies and
the stagnation zone is compressed, than during periods of low wind (compare Figs 15a and b). We also expect fewer particles to strike tall steps, with long stagnation zones, than short steps; this could place an upper limit on the heights of snow-steps.

    Finally, the erosion rate of solid materials is proportional to the energy of the impacting particles minus a threshold energy (Anderson, 1986). Therefore, steps can only be eroded if two conditions are met: the steps must be struck by saltating particles, and those particles must have more than the threshold energy upon impact. In Fig. 7b, we showed an example of differential
erosion on two layers of snow-steps. As these steps were subject to the same fluxes of wind and snow, we assume that the older, non-eroding layer had a higher erosion threshold. The threshold energy for the erosion of snow, unlike the erosion of bedrock, depends on time and weather. Dry, undisturbed snow sinters, or hardens, with time (Colbeck, 1998). Sintering can be accelerated by wind action (Colbeck, 1991), and is accelerated by several orders of magnitude by the presence of any liquid water (Blackford, 2007). Previous work at our field site found that snow-steps stop moving 5–7 days after snowfall, or
immediately after the temperature rises above $-1^{o}$C (Kochanski et al., 2018). Based on our current observations, however, we do not expect snow-step erosion rates to be determined by the average properties of the snow pack. Fig. 9c shows five snow-steps moving at different velocities, despite being mere meters apart. Those snow-steps decelerated over time. We infer from this that snow-step erosion rates are a function of the age of the step, but that this age differs from step to step across the shifting landscapes of bedform-covered snow.





## 4.4 Stealth dunes

We presented one previously undocumented erosional bedform: the stealth dune. Like other erosional bedforms, they have vertical windward edges, and like sastrugi they present points to the wind. Unlike sastrugi, however, they have distinct crescent shapes and are they are not arranged in a regular pattern. We saw stealth dunes only on the surface of Barker Reservoir, and

have only found one record of them in the prior literature: Cornish (1902) sketched an "erosion form analogous to a barchan" on land in British Columbia. Cornish hypothesized that these dunes are the eroded remnants of transverse waves. We concur. We occasionally see complete waves downwind of the stealth dunes on Barker Reservoir. Our observations allow three reasons why these dunes are rarely reported: (1) they are formed from snow waves, and little previous literature has documented snow-waves, (2) Barker Reservoir, with its narrow valley and upwind town, provides a rare combination of high, unidirectional winds

and low snow supply, or (3) stealth dunes are visible only in contrast to a dark surface, like lake ice.

## 4.5 Sastrugi formation

Sastrugi are the most widespread snow bedform (Filhol and Sturm, 2015), and can be the largest (Mather, 1962). They are therefore more interesting, from a broad view of the polar sciences, than other bedforms. We fear, however, that they are also more complex.

To develop a model of sastrugi evolution, we will need to overcome major field and computational challenges. First, stationary sastrugi geometries have not yet been characterized in detail. This problem may be solved by ongoing LiDAR studies. Second, sastrugi evolution is not easy to observe. Kochanski et al. (2018) found that sastrugi are formed during winds of at least 20 m/s (45 mph); in our study, we captured many hours of data without observing an instance of sastrugi formation. Third, sastrugi, unlike snow-steps, are fully three-dimensional features, and wind-blown snow follows winding three-dimensional paths

between their points. Moreover, we have shown that even a simpler bedform, the snow-step, is stabilized by complex flow structures and flow detachments. We therefore suspect that a successful model of sastrugi must resolve the three-dimensional flow structures around sastrugi points. We are, however, optimistic that grain sorting and wind-pumping are secondary effects that could be excluded from a useful model. Such a model, especially if it included the motion of blowing snow grains, would still require considerable computational expense.

## 4.6 Cycles of deposition and erosion

In Fig. 13d, we presented a conceptual model of a bedform in which aeolian transport, snow accretion, and erosion coexist. This is a mixed-surface bedform, in which loose snow and hardened snow alternate on the ground. The preeminent examples of mixed-surface bedforms are snow-waves. In the field, we saw wave crests made of loose, granular snow, with snow-steps forming in cohesive snow on their upwind sides (e.g. Fig. 8c). From these observations, we infer that loose snow is deposited

on the downwind lee of the wave, that the snow becomes cohesive in the time it takes for the wave to move past, and that some but not all of the newly-deposited snow is eroded into steps. At least some snow grains are thus accreted onto the surface during passage of the wave, some of which are then eroded back out of it with every passing wave (Fig. 12). If this conceptual model



is correct, then the nature of hardened-surface bedforms — and the presence or absence of accreted snow — is determined by the frequency of passing snow dunes and waves.

This pattern invites us to consider cycles of erosion and deposition on longer length and time scales. When a region of snow is deeply eroded, it releases snow particles into the wind. This increases the flux of transported snow into downwind regions,

pushing them from erosional to depositional regimes. Snow bedforms, then, are local manifestations of long-distance snow transport.

## 5   Conclusions

Here we have presented numerous examples of snow bedform movement to illustrate the modes of bedform growth and evolution. These examples are drawn from a library of over 1000 hours of time-lapse footage of snow bedform evolution,

available at Kochanski (2018b), and from detailed field observations in the Colorado Front Range. The data include a large number of observations of snow-waves (examples in § 3.1.6) and the first description of an erosive feature we term the 'stealth dune' (Fig. 10).

We have used the observations published here to develop conceptual models of the evolution of snow barchans, snow-steps, and snow waves. We propose that snow bedforms should be characterized in terms of the primary processes that form them:

snowfall, aeolian transport, erosion of cohesive substrates, and sintering.

These processes are all well-known to snow scientists and Earth surface scientists, but their interactions have not yet been studied. We hypothesize that future studies of snow bedforms will reveal new regimes of self-organization in nature, and lead us towards a quantitative understanding of the snow features that cover the alpine and polar regions of Earth.





*Data availability.* Time-lapse observations from March 2016–April 2017 are archived at http://doi.org/10.5281/zenodo.1253725 (Kochanski, 2018b). Weather (Losleben, 2018a) and precipitation (Losleben, 2018b) data for the Niwot Ridge field site are available from the Niwot Ridge Long-term Ecological Research program at niwot.colorado.edu.

*Author contributions.* K. Kochanski acquired funding for field equipment, carried out the field campaign, analysed the data and wrote the
manuscript. G. Tucker and R. Anderson supervised the project. All authors provided critical feedback and helped shape the research, analysis and manuscript.

*Competing interests.* The authors have declared that no competing interests exist.

*Acknowledgements.* This research was supported by a Department of Energy Computational Science Graduate Fellowship (DE-FG02-97ER25308) and by a University of Colorado Chancellor's Fellowship. Field equipment was funded by the American Alpine Club, the
Memorial Research Fund of the Colorado Scientific Society, and a Patterson Award from the University of Colorado Department of Geological Sciences. Field assistants were supported by the University of Colorado Undergraduate Research Opportunities Program. Logistical support and climate data were provided by the Niwot Ridge Long-Term Ecological Research Project and the University of Colorado Mountain Research station through NSF cooperative agreement DEB-1637686. We thank Irina Overeem, Clea Bertholet, Chelsea Herbertson, Madonna Yoder and Richard Barnes for help in the field. We extend additional thanks to Madonna Yoder and Chelsea Herbertson for data
analysis leading to the construction of Fig. 3.



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
