# Peer review of "The evolution of snow bedforms in the Colorado Front Range and the processes that shape them"

_The Cryosphere, 2018_

## Referee Comment (RC1) · Anonymous Referee #1 · 11 Mar 2019

This manuscript provides excellent physical insights into snow bedform dynamics, built on qualitative and quantitative observations of the snow surface evolution in the Colorado Front Range. The authors attempted a comprehensive exploration of the main factors influencing formation and evolution of a number of snow bedforms. I particularly appreciated the effort that the authors made to provide physical interpretations of their observations, discussing the complex interplay among snowfall, drifting snow, and sintering. The discussion session is enlightening with respect to the dynamics of snow step erosion and the similarities/differences between sand and snow dunes. Moreover, the manuscript seems to identify a new type of snow features, the stealth dune.

I think this is a valuable contribution to our understanding of snow bedform evolution that advances well beyond previous work. I recommend publication after the authors consider the comments below.

- Page 1, line 2: Maybe replace warmth with heat

- Page 1, line 8: the authors use the expression "retreating downwind" here and in other places. The verb retreat seems to me more suitable to describe an upwind movement. I would suggest to replace with advect to avoid confusion.

- Section 1.1: the authors differentiate between isolated dunes and close-packed dunes. However, the minimal distance at which dunes do not affect each others is not specified. I imagine that this distance is larger for dunes aligned with the wind direction, as the upwind dune can shed a wake several times its height. Conversely, side by side dunes can behave independently at reasonably short distance. Can the authors provide any estimations?

- Section 2.1: It would be useful to know the orientation of the ridge with respect to the main wind wind direction. This would tells us something more about the overall mass balance of the study site, such as the occurrence of snowfall preferential deposition and the relative importance of erosion and deposition.

- Section 2.2: Additional information on the camera setup would be useful. Was the camera looking perpendicular to the main wind direction? What was the elevation from the surface? What are the errors on the estimations caused by the distortion by distance?

- Page 6, line 6: "The falling snow that we observed often settled into plane beds". I imagine this was only the case for low wind speed conditions.

- Page 6, line 16: I think that self-similarity is not exactly what the authors show. Self-similarity would imply that the surface features look similar across different length scales, but the range of distances here seems to be quite narrow. You could simply say

that dune height and width scale linearly.

-Page 6, line 18: remove second 'of'.

- Page 6, line 26: It is not clear to me how you could track simultaneously blowing snow and dune evolution. Was the surface always visible from the camera during blowing snow? Or did you take manual measurements?

- Page 6, line 33: "Instead of the other way around". What do you mean exactly? It is hard to imagine how blowing snow fluxes could be affected by dune velocities.

- Figure 2, caption: Please mention that panel (b) only shows to close-packed dunes and (d) only isolated barchan dunes.

- Figure 3: Can you assign units to the axes? Approximate distances are better than no units.

- Page 8, line 13: the term "frequency" seems to indicate a cyclic process rather than a displacement. I think 'velocity' would be more appropriate. Can you assign approximate distance units based on camera pixels and/or distance from the camera?

- Page 8, line 18: What is your interpretation of the observation that sand-free ripples faded away? Is it related to larger sublimation?

- Figure 4c: Can you plot the correlation coefficient rather than the cross-correlation? It would help to evaluate how important is the correlation peak.

- Figure 9: I think you are expressing distances in too many different ways. Sometimes approximate distance units (figure 3), sometimes no units, sometimes fraction of image (here). The quality of the paper would improve if you could be consistent in the method used to quantify distances.

- Page 17, line 13: "snow dunes must somehow give way to snow waves and viceversa". From Figure 11 there seem to be no transitions from snow waves to snow dunes.

- Figure 11: In the text, you seem to describe region 2 and 3, but I didn't notice any reference to region 1 and 4. Why do you differentiate these two regions?

- Page 18, line 3: "The cyclic behavior is shown by the transition in region 3". From what you say earlier, all points laying below the 1:1 line are unidirectional. Then I think you are referring just to the part of region 3 above the 1:1 line in Figure 11?

- Page 19, line 5: Why do you list isolated barchan dunes in mixed-surface bedforms only? Did you not observe isolated dunes in loose-snow conditions? And viceversa, did you not observe close-packed dunes in mixed-surface conditions?

- Page 20, line 5: 'but much higher than the standard values for planar snow (0.5 mm)". 0.2 mm is not much higher than 0.5 mm, or am I missing something?

- Page 20, line 9-10: Readers may not be familiar with the interplay between shear velocity and settling velocity in setting the transport dynamics of particles. Please provide some references.

- Page 20, line 1-15: Overall, I find this paragraph not very well connected to the previous discussion. I suggest you clarify why this information is relevant for bedform dynamics.

- Page 20, line 22: rather than "time-dependent" I would say that it depends on snow properties such as grain size, grain shape, and sintering, which vary in time.

- Page 20, line 27: Can you clarify why suspension is relevant to dune evolution? If suspended particles do not interact with the surface, how can they influence the dunes?

- Page 21, line 3: How did you calculate the effective density?

- Page 21, line 30: Correct "grans" with "grains".

- Page 22, line 1: In high Reynolds number flows, the length of the recirculation zone should not be sensitive to the wind speed. Can you provide any reference to previous studies that showed this?

- Page 23, line 7: Similarly to my previous comment, I'm not sure that the length of the stagnation zone changes with the wind speed - assuming the flow is Reynolds number independent, as I assume may be the case here. Please provide some additional explanation or references.

- Page 23, line 12: What is this threshold energy? Previous studies suggested that this threshold energy is that necessary to break cohesive bonds (e.g., Comola and Leaning 2017, Gauer 2001). So this threshold depends on the conditions of the snow surface.

---

## Short Comment (SC1) · 14 Mar 2019

Dear Reviewer 1,

Thank you for your comments.

I am working on incorporating your suggestions into our final manuscript, but would like to address the most technical issues early to allow for additional feedback.

The first draft of this manuscript, I assumed that changing the flow speed would change the lengths of the recirculation zones around dunes and snow-steps. Your comments on sections 4.2 and 4.3 of this draft questioned this assumption. To address your

question, I worked with colleague Aaron Hurst to address this question with a series of short numerical experiments in which we resolved the flow around roughly-shaped dunes and snow-steps using ComSOL, a fluid dynamics solver. As you suggested, we found that the length of the recirculation zone around the snow-step and the snow dune did not change significantly over the range of wind speeds experienced at our field site.

I have adjusted Figures 14 and 15 to reflect this change; the new versions are attached below for your review. Before the final version of the paper I will adjust the text explain wind-speed-dependent bedform behaviors in terms of armoring, saltation hop lengths, and grain energy rather than flow structure.

Please let us know if you have any further comments, or see any non-physical behavior in the updated figures.

Yours, Kelly Kochanski

———————————————————

[Figure]

**Fig. 1.**

[Figure]

**(a) Slow flow**

low pressure, few impacts,
no loose grains

gentler impacts
loose grains may accumulate

**(b) Fast flow**

low pressure, few impacts,
no loose grains

high pressure, energertic impacts
loose grains blown away immediately

**Fig. 2.**

---

## Referee Comment (RC2) · Anonymous Referee #2 · 25 Mar 2019

General comments:

This study uses an interesting set of pictures of the snow surface collected in a mountainous location to propose a comprehensive and original synthesis on snow bedform dynamics that undoubtedly contribute to improve our understanding on this quite undocumented subject. The manuscript is really well written, the semantics is appropriate and I really enjoyed reading it. The various and pertinent descriptions made from analysis of the footage and field measurement as well as the efforts put to physically relate and interpret them in terms of driving processes is really appreciable. Ways forward are provided and remaining gaps are identified. I recommend the paper as suitable for

publication providing the authors can address the following minor comments.

- P6, L8: "unsintered, unbroken snowflakes": Strictly speaking this sounds like idealized conditions since sintering starts naturally with vapour transfer as soon as snow is deposited and overburden pressure causes breaking of original crystal forms. Prefer simply loose snow or fresh snow layers.

- P6, L32-33: "Instead of the other way round": Do you mean that your conclusion, which is self-sufficient and quite relevant to me, is less intuitive than the reversal involving an influence of dune velocities on blowing snow fluxes? Because it is actually the opposite.

- P18, L13: "In the aeolian world": such a seductive phrase. I know rules are sometimes meant to be broken, but yet this is not suited for a scientific paper I'm afraid. Stop the sentence after "analogous" or replace with "among aeolian features"? I'm just suggesting.

- P20, L07 and elsewhere: Prefer "friction velocity" to "shear velocity".

- P20, L4-6: Comparatively to the wide range of z0 values that have been reported for aerodynamically rough surfaces, 0.2 mm is not "much lower" than 1 mm. See for instance Jackson and Carroll (1978) who reported centimetric z0 values for winds blowing perpendicularly to the sidewalls of high sastrugi (there seems to be a confusion in the actual value of z0 for planar snow since 0.5 mm is higher than 0.2 mm). Note that changes in z0 of several orders of magnitude can also occurs depending on small shifts in wind direction without changes in wind speed, which may involve a few comments on the wind directional range at your study site. But, in line with my next comment, I don't see the added-value of this section.

- P20, L1-15: A bit of confusion here. Friction velocity intervenes for the lifting of particles of the surface and must overcome the cohesive and gravitational forces to trigger saltation. Once airborne, suspension of particles is ensured by a (wind) drag

force high enough to compensate for the gravitational pull. Moreover, there is no need to my opinion for such approximative calculations to finally state that smaller and lighter particles are preferably carried out by suspension than larger ones. Just evoking that the suspension transport mode is governed by a local dynamical balance between the downward gravitational force and the upward drag force due to turbulence logically permits such a statement without any quantitative illustration. From this perspective the estimation of the friction velocity in the above paragraph is not needed anymore. In addition I don't see clearly the link of this paragraph with the rest of the text. Maybe consider removing it.

- P21,L3: Could you give references to support the values attributed to the effective density and diameter of saltating snow grains?

- P21,L27: How do you know? Any reference?

- P21, L27-31: That final part of the discussion is not really convincing and does not shed light on anything. As you can't provide any measurements of grain size there is no need to speculate, even qualitatively, on suspension rates. You could remove it without altering the quality of the discussion, which is already quite long.

- P25, L5: This generalization sounds a bit hasty. This is not necessarily true in windswept regions subject to quasi-unidirectional flows and relatively high snowfall rates and where erosional bedforms prevail, such as crest and/or windward slopes in mountainous regions, the accumulation zone of the Greenland ice sheet or a large portion of the Antarctic coast. That is, in many regions. Be more specific on the conditions required for your assertion to hold true.

Typo and misspelling: - P1, L20: You have reversed last and first names: "Bellot" is Hervé's last name. - P1,L20: Spell Naaim-Bouvet instead of Naiim. - P6, L32: "blowing snow" instead of "blowing slow" - P10, L12: I guess "collected" must be used instead of "collect" - P10, L12: "continuously" instead of "continuosly" - P21, L30: "grains" instead of "grans" - P22, L3: "a too large number" instead of "too large a number" - P24, L4:

"are they" must be removed

---

## Author Comment (AC1) · 29 Mar 2019

Dear reviewers and Dr. Schneebeli,

The final author response is attached as a supplementary pdf.

Comments from reviewer #1 are addressed in p1-8. Comments from reviewer #2 are addressed on p9-11, and the revised manuscript is included in p12-end. All changes are marked in the reviewer comments.

Please also note the supplement to this comment:

[Figure]

https://www.the-cryosphere-discuss.net/tc-2018-293/tc-2018-293-AC1-supplement.pdf

---

## Author Response (AR1)

**Author's response to Reviewer #1**

**Review:**
This manuscript provides excellent physical insights into snow bedform dynamics, built on qualitative and quantitative observations of the snow surface evolution in the Colorado Front Range. The authors attempted a comprehensive exploration of the main factors influencing formation and evolution of a number of snow bedforms. I particularly appreciated the effort that the authors made to provide physical interpretations of their observations, discussing the complex interplay among snowfall, drifting snow, and sintering. The discussion session is enlightening with respect to the dynamics of snow step erosion and the similarities/differences between sand and snow dunes. Moreover, the manuscript seems to identify a new type of snow features, the stealth dune.

I think this is a valuable contribution to our understanding of snow bedform evolution that advances well beyond previous work. I recommend publication after the authors consider the comments below.

**Author's comments**
Thank you for the supportive review, and for the constructive comments included below.

We included a short response to your most technical comments concerning Figs. 14 and 15 in the interactive discussion. To recap, you pointed out that the wind speeds at our field site were too to create significant compression in the air. We confirmed that you were correct, and we adapted Figs. 14 and 15 to show that the lengths of the recirculation zones are constant with wind speed, as shown below:

[Figure]

**Line-by-line comments**
Page 1, line 2: Maybe replace warmth with heat-  *Done.*

Page 1, line 8: the authors use the expression "retreating downwind" here and in other places. The verb retreat seems to me more suitable to describe an upwind movement. I would suggest to replace with advect to avoid confusion.-

This is a good point. We hoped that 'retreat' would imply mass loss and movement away from the force of the wind, but we can see that it may also confuse some readers who see 'retreat' as a movement opposite to the wind.
'Advect is a good suggestion', but it usually refers to an object that travels without changes to its mass or shape. Since we are describing a process of erosion, this might be misleading.

We have replaced 'retreat' with **'migrate [downwind]'**, which we hope will encourage readers to think of motion in the shape of the step, not in the mass that forms it.

Section 1.1: the authors differentiate between isolated dunes and close-packed dunes. However, the minimal distance at which dunes do not affect each others is not specified. I imagine that this distance is larger for dunes aligned with the wind direction, as the upwind dune can shed a wake several times its height. Conversely,side by side dunes can behave independently at reasonably short distance. Can the authors provide any estimations?-

We believe that the important distinction here is not the distance between dunes but the fraction of the surface which is covered by fresh snow. We have amended the text in Section 1.1 to make this clear:

> Barchan dunes are crescent-shaped, or two-horned, dunes. They have well-defined crests, with a gentle upwind slope and a steep downwind slope that curves into two forward-pointing arms \citep{Filhol2015, Petrich2012, Doumani1967, Kobayashi1980, Goodwin1986}.
> **In this paper, `barchan dune' (or `isolated dune') refers specifically to crescent dunes that do not cover the surface completely, but expose inter-dune areas of old snow, bare ground, or ice.**

> Close-packed dunes  have defined crests. Unlike isolated barchans, they cover the surface completely.
> **We distinguish close-packed dunes from isolated barchans because they have different thermal properties and we expect them to follow different evolutionary trajectories.** First, fresh snow is more reflective and less thermally conductive than old snow, ice, or bare ground. Even small gaps in the snow cover create a positive feedback cycle that leads to rapid snowmelt in warm or sunny weather \citep{Petrich2012}. Second, fresh snow is usually less dense, and more erodible than old snow, ice, or dirt. Thus, exposed inter-dune areas make the snow surface inhomogeneous, which makes the dynamics of bedform evolution considerably more complex. We discuss this in \textsection 4 and \textsection 4.6.

Section 2.1: It would be useful to know the orientation of the ridge with respect to themain wind wind direction. This would tells us something more about the overall mass balance of the study site, such as the occurrence of snowfall preferential deposition and the relative importance of erosion and deposition.-

Good catch. Amended text:
> In winter the wind on Niwot Ridge blows almost without exception from the west-northwest, driven by temperature gradients over the Continental Divide. **The ridge also runs east-southeast to west-northwest, parallel to the prevailing wind. The saddle site is downwind of a gentle bump in the ridge, and collects deep snow during the winter although other parts of the ridge are occasionally scoured free of snow.**

Section 2.2: Additional information on the camera setup would be useful. Was the camera looking perpendicular to the main wind direction? What was the elevation from the surface? What are the errors on the estimations caused by the distortion by distance?-

Added:
>**The cameras were angled downwind or perpendicular to the prevailing wind to prevent snow from covering the lens; we indicate the wind direction for each image.**
>**The cameras were mounted on 2 m tripods. The distance between the cameras and the surface decreased as snow accumulated; we reset them on the surface once a month.**
>**The uncertainty in the camera elevation makes it difficult for us to estimate distance directly from the images.** We **therefore** measured bedform sizes and velocities both in-person, using meter sticks, and by reference to poles with 10-cm stripes placed in front of the cameras. **Where direct measurements are missing,** for example because the poles blew away, we convey sizes, speeds and intensities in non-dimensional units.

Page 6, line 6: "The falling snow that we observed often settled into plane beds". I imagine this was only the case for low wind speed conditions.-

Amended:
>The falling snow that we observed often **fell in still air** and settled into plane beds.

Page 6, line 16: I think that self-similarity is not exactly what the authors show. Self-similarity would imply that the surface features look similar across different length scales, but the range of distances here seems to be quite narrow. You could simply say that dune height and width scale linearly.-

Changed to:
>The dune heights and widths **scale linearly.**

Page 6, line 18: remove second 'of'.-
Done.

Page 6, line 26: It is not clear to me how you could track simultaneously blowing snow and dune evolution. Was the surface always visible from the camera during blowing snow? Or did you take manual measurements?-

The dune in this case was only about 2 m from the camera. Therefore, the dunes were clearly visible through the snow in perhaps 98% of the frames of the video. We included the footage in the supplementary video (Video S1, starting at 1:07); if you watch the footage both blowing snow and dune movement are quite clearly visible.

Added:
>**The dune was clearly visible through the snow in all but a handful of frames (see video S1 at 1:07).**

Page 6, line 33: "Instead of the other way around". What do you mean exactly? It is hard to imagine how blowing snow fluxes could be affected by dune velocities.-

You are correct. We removed confusing phrase:
>""

Figure 2, caption: Please mention that panel (b) only shows to close-packed dunes and (d) only isolated barchan dunes.-

Added:
        b) Time-lapse of **close-packed** dunes [...]
        d) Time-lapse of i**solated** barchan dunes [...]

Figure 3: Can you assign units to the axes? Approximate distances are better than no units.-

For clarity, we have relabeled the axis with a unit 'L' that is approximately equal to 1 m. We have changed other figures that lack exact distances, such as Figure 9, to match. We hope that this will make things easier for readers while still accurately expressing our uncertainty in our estimates of the distances in these figures.

As the reviewer's comment on Section 2.2 drew our attention to need to carefully address the uncertainties in our distance measurements, we hope that you will agree that we have chosen a reasonable presentation given the (unfortunately incomplete) distance data we have.

 Page 8, line 13: the term "frequency" seems to indicate a cyclic process rather than a displacement. I think 'velocity' would be more appropriate. Can you assign approximate distance units based on camera pixels and/or distance from the camera?-

We converted the units from frequency to time for clarity, and added an approximate velocity based on approximate distance units.
        The crests advanced  **by one wavelength every $33 \pm 3$ s. Although scale bars were missing from this image, if the ripples had typical 10–25 cm wavelengths, they would move at 11–27 m/hr.**

These high velocities are consistent with our other observations of ripples.

Page 8, line 18: What is your interpretation of the observation that sand-free ripples faded away? Is it related to larger sublimation?-

I suspect that the wind was blowing too fast for snow to be stable in bedload, and that the snow that was not armored by heavier sand grains blew away in suspension. Perhaps the sand-free ripples had lost their sand grains only recently and were in the process of blowing away.

Figure 4c: Can you plot the correlation coefficient rather than the cross-correlation? It would help to evaluate how important is the correlation peak.-

We use this plot to find the lag between the pulse of snow and the increase in dune velocity. Cross-correlations are a standard method for finding the lag between two signals, and so they are appropriate and relevant here.

Figure 9: I think you are expressing distances in too many different ways. Sometimes approximate distance units (figure 3), sometimes no units, sometimes fraction of image(here). The quality of the paper would improve if you could be consistent in the method used to quantify distances.

We changed the figure in question to be consistent with Figure 3. The paper now uses two self-consistent distance units: metres when distances are known with high confidence, and 'L'

when real distances are not known, where 'L' is an arbitrary unit that equals one metre plus or minus some uncertainty listed in the figure caption.

This increases the consistency of the distance expressions, while endeavoring to make the systematic uncertainties in our distance measurements clear to readers.

Page 17, line 13: "snow dunes must somehow give way to snow waves and vicev-ersa". From Figure 11 there seem to be no transitions from snow waves to snow dunes.

Good catch. Amended to:
    "there [must be] path for **isolated barchans** to transition into snow waves and vice versa."

Figure 11: In the text, you seem to describe region 2 and 3, but I didn't notice anyreference to region 1 and 4. Why do you differentiate these two regions?-

These regions were drawn to be consistent with the loose-snow bedform, hardened-snow bedform, and mixed-surface bedforms categories that we use in the discussion. We have now made the motivation for the region-grouping clear in the text:
    "We have grouped together transitions that share similar characteristics. Transitions
    that involve a surface made of entirely loose snow are in Region I (yellow). Transitions
    from one type of hardened, erosional surface to another are in Region II (orange).
    Transitions between surfaces that expose a mix of loose snow and hardened snow,
    such as waves, are in Region (pink), and transitions in which hardened surfaces turn
    into mixed surfaces, or vice-versa, are in Region III (red). These groups of bedforms are
    discussed further in \textsection 4."

We have also re-organized our discussion of Figure 11 so that all four regions are mentioned in order:
    "The transitions in **region I** and **region II** appear to be irreversible [...]
    [...] the transitions in **region III** appear to drive cyclic evolution trajectories. [...]
    [...] we did not directly observe any transitions in **region IV** [...] "

Page 18, line 3: "The cyclic behavior is shown by the transition in region 3". Fromwhat you say earlier, all points laying below the 1:1 line are unidirectional. Then I think you are referring just to the part of region 3 above the 1:1 line in Figure 11?-

In mathematical terms, cyclicity is a property of an entire graph or matrix, not a particular transition within a graph.

For example, the transition from snow-waves to isolated barchans is currently above the 1:1 line. But, if I had decided to list the the snow-waves before isolated barchans on the graph, the snow-wave to barchan transition would fall below the 1:1 line, and instead the transition from isolated barchans to snow-waves, which is currently below the 1:1 line, would be above it.

I have amended the text to make this clearer, and to encourage readers to look for evidence of cyclicity in Fig 12 (where it is shown graphically) instead of Fig. 11 (where the same information is shown in matrix form, which is less intuitive):

    The cyclic transitions are shown graphically in Fig. 12.  This graph contains the same
    information that is represented in tabular or matrix form in regions II, III, and IV. If
    bedform evolution were unidirectional, we would be able to arrange Fig. 11 such that
    all transitions lay on **the same side** of the 1:1 line, **or arrange Fig. 12 without any
    loops or repetition.**

Page 19, line 5: Why do you list isolated barchan dunes in mixed-surface bedforms only? Did you not observe isolated dunes in loose-snow conditions? And vice-versa, did you not observe close-packed dunes in mixed-surface conditions?-

This should now be clarified in the text by the updated definitions of isolated and close-packed dunes (see response to comment on section 1.1). We've updated the definition of isolated dunes to be dunes with exposed surfaces beneath them (old snow, bare ground, etc), so these dunes are mixed-surface features by definition.

Page 20, line 5: 'but much higher than the standard values for planar snow (0.5 mm)".0.2 mm is not much higher than 0.5 mm, or am I missing something?-

Good catch. That was meant to be 0.05 mm. Fixed.

Page 20, line 9-10: Readers may not be familiar with the interplay between shear velocity and settling velocity in setting the transport dynamics of particles. Please provide some references.-
Page 20, line 1-15: Overall, I find this paragraph not very well connected to the previous discussion. I suggest you clarify why this information is relevant for bedformdynamics.-

In response to this and a similar comment from Reviewer 2, we have restructured this paragraph so that it focuses clearly on the bedform-shaping processes in Fig. 13. We have also removed some of the less-well connected information, including the descriptions of Rouse numbers and shear velocities. This shortened the discussion by 300 words.

The section now reads:
Fig.~13 illustrates the major snow processes that shape bedforms: snowfall, aeolian transport, erosion, and sintering. Here, we analyse the relative importance of these fluxes as a function of snow grain size and wind speed.

Fig.~13a shows a wind too weak to move any snow grains. This occurs when the force that the wind exerts on the surface is insufficient to overcome gravity and friction and lift any grains.

Fig.~13b shows loose-snow bedforms created by horizontal snow transport. We expect these bedforms to be created when the wind friction velocity is high enough to mobilize snow, but but not so high that all the snow is lifted away from the surface and into suspension. \citet{Li1997} found that dry snow is mobilized by winds higher than 7--14 m/s, measured at 10 m elevation. \citet{Clifton2006} found slightly higher thresholds that increased with particle density and size.

Fig.~13 shows a hardened snow surface being eroded. The erosion rate of a solid surface is proportional to the energy of the impacting particles, minus a threshold energy that depends on the material hardness \citep{Anderson1986a}. Thus, erosion requires a supply of loose, high-speed snow particles.
The threshold erosional energy of snow is not well documented in the literature, as most snow gets harder over time. This process, known as sintering, is accelerated slightly by humidity, warmth \citep{Colbeck1998}, small grain sizes, and wind action \citep{Colbeck1991}, and accelerated by orders of magnitude by the presence of liquid water \citep{Blackford2007}.

Finally, Fig.~13d shows a surface in which sintering, transport, and erosion happen at comparable rates. The resulting bedforms are discussed in section 4.6.

Page 20, line 22: rather than "time-dependent" I would say that it depends on snowproperties such as grain size, grain shape, and sintering, which vary in time.-

Clarified:
> Most snow gets harder over time. This process, known as sintering, is accelerated slightly by humidity, warmth \citep{Colbeck1998}, small grain sizes, and wind action \citep{Colbeck1991}, and accelerated by orders of magnitude by the presence of liquid water \citep{Blackford2007}.

Page 20, line 27: Can you clarify why suspension is relevant to dune evolution? If suspended particles do not interact with the surface, how can they influence the dunes?

Particles can move from saltation to suspension either because the wind rises, or the particles become smaller due to sublimation or fragmentation.

Thus, under conditions of rapid sublimation or wind acceleration, the dunes will continuously lose mass as particles move into sublimation and cease interacting with the surface. This is a transient scenario that has not been discussed in detail in the sand literature, but is likely to have a significant impact on snow dune shape.

As part of the reorganization of section 4.1, we now explain this in the paragraph starting:
> Fig. ? shows loose-snow bedforms being created by horizontal wind-driven snow transport. We expect these bedforms to be created when the wind shear velocity is high enough to mobilize snow [...]

Page 21, line 3: How did you calculate the effective density?-
The effective values for snow grain density vary widely. Snowpacks usually have a density of 100-300 kg/m^3.

Page 21, line 30: Correct "grans" with "grains".- Fixed.

Page 22, line 1: In high Reynolds number flows, the length of the recirculation zone should not be sensitive to the wind speed. Can you provide any reference to previous studies that showed this?

You are correct, thank you for catching this error in our conceptual model. We worked with a colleague to check this using a fluid dynamics solver, Comsol, and we did not find any evidence that the length of the recirculation zone changed significantly at the pressures and wind speeds we expect to see at our field site.

We have updated Figure 14 as shown at the top of this review, and now note that the change in dune behavior between wind speeds will depend entirely on the saltation hop length of the grains, not the recirculation zone.

Page 23, line 7: Similarly to my previous comment, I'm not sure that the length of the stagnation zone changes with the wind speed - assuming the flow is Reynolds number independent, as I assume may be the case here. Please provide some additionalexplanation or references.-

You are again correct. We have updated Figure 15 as shown at the top of this review, and now explain the effect of wind speed on step erosion solely in terms of particle energy and armoring by loose grains.

Page 23, line 12: What is this threshold energy? Previous studies suggested that this threshold energy is that necessary to break cohesive bonds (e.g., Comola and Leaning2017, Gauer 2001). So this threshold depends on the conditions of the snow surface.

Added and re-organized the following discussion:

**The threshold erosional energy of snow is not well documented in the literature. Most snow gets harder over time. This process, known as sintering, is accelerated slightly by humidity, warmth \citep{Colbeck1998}, small grain sizes, and wind action \citep{Colbeck1991}, and accelerated by orders of magnitude by the presence of liquid water \citep{Blackford2007}.**

We do not think that estimating a numerical value for this threshold energy in snow is within the scope of this paper, as we are certain that this value would vary in both space and time at our field site.

**Author's response to Reviewer #2**

**Review:**
This study uses an interesting set of pictures of the snow surface collected in a mountainous location to propose a comprehensive and original synthesis on snow bedform dynamics that undoubtedly contribute to improve our understanding on this quite undocumented subject. The manuscript is really well written, the semantics is appropriate and I really enjoyed reading it. The various and pertinent descriptions made from analysis of the footage and field measurement as well as the efforts put to physically relate and interpret them in terms of driving processes is really appreciable. Ways forward are provided and remaining gaps are identified. I recommend the paper as suitable for publication providing the authors can address the following minor comments.-

**Author's comments:**
Thank you for your supportive review. We're glad to hear that paper was an enjoyable read.

We appreciate the attention to detail you have given to our writing and references, and we have corrected all of the minor errors (typos, mis-spelled references, and unclear phrasing) that you identified.

In response to your most technical comments (on discussion paper P20 L1-15, the discussion of Rouse numbers and friction velocities), we have removed our estimates of numbers we could not measure in the field, as well as the introduction to friction velocities and Rouse numbers. This shortened the discussion by 300 words.

The section now reads:

Fig.~13 illustrates the major snow processes that shape bedforms: snowfall, aeolian transport, erosion, and sintering. Here, we analyse the relative importance of these fluxes as a function of snow grain size and wind speed.

Fig.~13a shows a wind too weak to move any snow grains. This occurs when the force that the wind exerts on the surface is insufficient to overcome gravity and friction and lift any grains.

Fig.~13b shows loose-snow bedforms created by horizontal snow transport. We expect these bedforms to be created when the wind friction velocity is high enough to mobilize snow, but but not so high that all the snow is lifted away from the surface and into suspension. \citet{Li1997} found that dry snow is mobilized by winds higher than 7--14 m/s, measured at 10 m elevation. \citet{Clifton2006} found slightly higher thresholds that increased with particle density and size.

Fig.~13 shows a hardened snow surface being eroded. The erosion rate of a solid surface is proportional to the energy of the impacting particles, minus a threshold energy that depends on the material hardness \citep{Anderson1986a}. Thus, erosion requires a supply of loose, high-speed snow particles.
The threshold erosional energy of snow is not well documented in the literature, as most snow gets harder over time. This process, known as sintering, is accelerated slightly by humidity, warmth \citep{Colbeck1998}, small grain sizes, and wind action \citep{Colbeck1991}, and accelerated by orders of magnitude by the presence of liquid water \citep{Blackford2007}.

Finally, Fig.~13d shows a surface in which sintering, transport, and erosion happen at comparable rates. The resulting bedforms are discussed in section 4.6.

**Line-by-line responses:**
P6, L8: "unsintered, unbroken snowflakes": Strictly speaking this sounds like idealized conditions since sintering starts naturally with vapour transfer as soon as snow is deposited and overburden pressure causes breaking of original crystal forms. Prefer simply loose snow or fresh snow layers.-
Reworded:
"They are the only surface type made from fresh snowflakes that have not saltated or broken during saltation"

P6, L32-33: "Instead of the other way round": Do you mean that your conclusion,which is self-sufficient and quite relevant to me, is less intuitive than the reversal involving an influence of dune velocities on blowing snow fluxes? Because it is actually the opposite.-
Your intuition is intriguing. We have added the following observation:
This positive correlation and positive lag is evidence of causation: high **incoming** fluxes of blowing snow drive high dune velocities . **Moreover, the dune movement must have contributed to the flux of blowing snow, because the dunes lost height during the observation. These blowing snow events appear to have create a positive feedback that turned snow dunes into wind-blown snow.**

P18, L13: "In the aeolian world": such a seductive phrase. I know rules are sometimes meant to be broken, but yet this is not suited for a scientific paper I'm afraid. Stop the sentence after "analogous" or replace with "among aeolian features"? I'm just suggesting.-
Thank you. Some unsuitable phrases are too pretty to remove before review.
Changed to:
"Other snow bedforms, such as snow-waves, are **not obviously analogous to other aeolian features**"

P20, L07 and elsewhere: Prefer "friction velocity" to "shear velocity".-
Sure. Changed all instances of "shear velocity" to "friction velocity". Added a note at first use of the phrases that "u_* is the friction velocity (sometimes called the 'shear velocity')".

P20, L4-6: Comparatively to the wide range of z0 values that have been reported for aerodynamically rough surfaces, 0.2 mm is not "much lower" than 1 mm.
Removed "much lower" comparison; now using only quoted values without evaluation.

See for instance Jackson and Carroll (1978) who reported centimetric z0 values for winds blowing perpendicularly to the sidewalls of high sastrugi (there seems to be a confusion in the actual value of z0 for planar snow since 0.5 mm is higher than 0.2 mm).
This is a typo; we have now corrected "0.5" to "**0.05**".

Note that changes in z0 of several orders of magnitude can also occurs depending on small shifts in wind direction without changes in wind speed, which may involve a few comments on the wind directional range at your study site. But, in line with my next comment, I don't see the added-value of this section.-
Following this comment, we have reworked this entire section to fit better within the text.

P20, L1-15: A bit of confusion here. Friction velocity intervenes for the lifting of particles of the surface and must overcome the cohesive and gravitational forces to trigger saltation. Once airborne, suspension of particles is ensured by a (wind) drag force high enough to compensate for the gravitational pull.
Moreover, there is no need to my opinion for such approximative calculations to finally state that smaller and lighter particles are preferably carried out by suspension than larger ones. Just evoking that the suspension transport mode is governed by a local dynamical balance between the downward gravitational force and the upward drag force due to turbulence logically permits such a statement without any quantitative illustration.

From this perspective the estimation of the friction velocity in the above paragraph is not needed anymore. In addition I don't see clearly the link of this paragraph with the rest of the text. Maybe consider removing it.-
P21,L3: Could you give references to support the values attributed to the effective density and diameter of saltating snow grains?-
We rewrote the section that sparked all of these comments. The new section was quoted at the top of this review.

P21,L27: How do you know? Any reference?-
Added references:
   "we know that dunes do not grow larger [than tens of centimeters] even in very cold locations, such as Antarctica (**Doumani, 1967)** and Alaska (**Filhol, 2015)**"

P21, L27-31: That final part of the discussion is not really convincing and does not shed light on anything. As you can't provide any measurements of grain size there is no need to speculate, even qualitatively, on suspension rates. You could remove it without altering the quality of the discussion, which is already quite long.-

Good point. Thank you for pointing out an opportunity for concision. We removed:

   ~~The Rouse number calculations formed at the start of \textsection~\ref{sec: discussion} indicate that grains >0.1 mm are consistently transported in bedload; in our experience, grains of this size and larger are very common in dunes and bedload, but without quantitative measurements of grain size we cannot constrain the rate of change of typical grain sizes through time.~~

P25, L5: This generalization sounds a bit hasty. This is not necessarily true in windswept regions subject to quasi-unidirectional flows and relatively high snowfall rates and where erosional bedforms prevail, such as crest and/or windward slopes in mountainous regions, the accumulation zone of the Greenland ice sheet or a large portion of the Antarctic coast. That is, in many regions. Be more specific on the conditions required for your assertion to hold true. Typo and misspelling: -
Good catch. We are used to thinking in terms of the net-depositional environment of our field site. We have reworded the paragraph to explain the range of possible effects:
   "When a region of snow erodes, it releases snow particles into the wind.
   Thus, the erosion of upwind bedforms may provide a snow flux that drives the evolution of downwind bedforms. **The effect of this snow flux will depend on the wind speed and the nature of the downwind bedforms. If the wind speed decreases, we expect blowing snow to be deposited in dunes, waves, or smooth drifts. In windswept conditions, we have seen blowing snow grains erode snow-steps, and remove mass from snow dunes."**

We corrected all the following typos:
P2, L20: You have reversed last and first names: "Bellot" is Hervé's last name. -
P2,L20: Spell Naaim-Bouvet instead of Naiim. -
P6, L32: "blowing snow" instead of "blowing slow" -
P10, L12: I guess "collected" must be used instead of"collect" -
P10, L12: "continuously" instead of "continuosly" -
P21, L30: "grains" insteadof "grans" -
P22, L3: "a too large number" instead of "too large a number" - Used simpler "too many"
P24, L4:C3"are they" must be removed

'

[revised manuscript text omitted]